# Cell-autonomous GABAARs are essential for NMDAR-mediated synaptic transmission, LTP, and spatial memory

Jing-jing Duan [1,2,3✉], Bin Jiang [3], Wei Yin[4], Yuan Lin [2,5], Guang-mei Yan[5] & Wei Lu [6,7]

## Abstract

GABA$_A$ receptors (GABA$_A$Rs) mediate most synaptic inhibition in the brain, but their cell-autonomous role in regulating glutamatergic transmission remains poorly understood. By targeting GABA$_A$R β1–3 subunit alleles (*GABRB1-3*) at once, we genetically eliminated GABA$_A$Rs in individual hippocampal CA1 pyramidal neurons. We find that single-cell silencing of GABAergic transmission does not alter AMPAR-mediated synaptic transmission, but leads to a reduction in NMDAR-mediated synaptic transmission, loss of long-term potentiation (LTP), and impaired spatial memory. Genetic rescue experiments reveal that NMDAR-mediated whole-cell currents and synaptic transmission depend on specific GABA$_A$R subtypes and are tightly regulated by neuronal excitability. Pharmacologically restoring NMDAR function in β123-CRISPR mice rescues both LTP and spatial memory deficits induced by the loss of GABA$_A$Rs in CA1 neurons. Our data uncover a previously unknown regulation of synaptic NMDAR functions by GABA$_A$Rs at the single-cell level and provide insight into excitation and inhibition balance between GABA$_A$Rs and NMDARs in the brain.

**Keywords** GABAARs; NMDARs; E/I Balance; LTP; Spatial Memory
**Subject Category** Neuroscience

## Introduction

The vast majority of fast inhibitory transmission in the brain is mediated by GABA acting on GABA$_A$ receptors (GABA$_A$Rs) (Jacob et al, 2008). GABA$_A$Rs are ligand-gated hetero-pentameric anion channels assembled from various combinations of 19 subunits: α (1–6), β (1–3), γ (1–3), δ, ε, θ, π, and ρ (Jacob et al, 2008; Duan et al, 2019; Nguyen and Nicoll, 2018). Most GABA$_A$Rs in the brain consist of two α subunits, two β subunits, and one γ or δ subunit (Jacob et al, 2008). Recent studies employed the CRISPR-Cas9 technique have shown that knockout (KO) of all three β subunits (β1–3) eliminates GABA$_A$R-mediated currents in hippocampal neurons, providing genetic evidence that a functional GABA$_A$R requires inclusion of the β subunit (Duan et al, 2019; Nguyen and Nicoll, 2018).

Due to the enormous diversity and complexity of GABA$_A$Rs, the native role of GABA$_A$Rs in sculpting neuronal structure and function has been inferred mainly by utilizing pharmacological blockade (Ben-Ari et al, 2007; Olsen and Sieghart, 2008). However, the global blockade of GABA$_A$R-mediated synaptic transmission by pharmacological approaches cannot distinguish between the cell-autonomous function of inhibitory synaptic transmission and the indirect effects on network activity (Lu et al, 2013). It may also eliminate competition among neurons, which can obscure the potential roles of inhibitory synaptic transmission in the regulation of neuronal structure and function (Buffelli et al, 2003). In addition, while global KOs of several different GABA$_A$R subunits have been generated and characterized (Rudolph and Möhler, 2004), progress in understanding the cell-autonomous role of GABA$_A$Rs in regulating synaptic function has been limited.

Excitation-inhibition (E/I) balance is critical for normal brain development and function (Turrigiano, 2012) and dysregulation of E/I balance has been implicated in several brain disorders (Sohal and Rubenstein, 2019). At the single cell level, it has been shown that excitatory glutamate receptors play a significant role in regulating GABA$_A$R trafficking and inhibitory transmission (Chiu et al, 2018; Horn and Nicoll, 2018; Lu et al, 2013; Wu et al, 2021). However, the cell-autonomous role of GABA$_A$Rs in the regulation of excitatory synaptic transmission remains largely unclear. AMPA receptors (AMPARs) and NMDA receptors (NMDARs) mediate the vast majority of fast excitatory neurotransmission in the brain. Recent studies have revealed that reduction or loss of GABA$_A$R-mediated inhibitory postsynaptic currents (IPSCs) in neurons have little effect on AMPAR-mediated excitatory postsynaptic currents (EPSCs) (Davenport et al, 2017; Fritschy et al, 2006; Gross et al, 2016; Han et al 2019; Patrizi et al, 2008; Uezu

[1]Department of Anatomy and Neurobiology, Zhongshan School of Medicine, Sun Yat-sen University, 510080 Guangzhou, China. [2]Advanced Medical Technology Center, The First Affiliated Hospital, Sun Yat-sen University, 510080 Guangzhou, China. [3]Guangdong Province Key Laboratory of Brain Function and Disease, Zhongshan School of Medicine, Sun Yat-sen University, 510080 Guangzhou, China. [4]Department of Biochemistry, Zhongshan School of Medicine, Sun Yat-sen University, 510080 Guangzhou, China. [5]Department of Pharmacology, Zhongshan School of Medicine, Sun Yat-sen University, 510080 Guangzhou, China. [6]Shenzhen Medical Academy of Research and Translation, 518132 Shenzhen, China. [7]Section of Synapses and Neural Circuits, National Institute of Neurological Disorders and Stroke, National Institutes of Health, Bethesda, MD 20892, USA. ✉E-mail: duanjj2@mail.sysu.edu.cn

et al, 2016; Yamasaki et al, 2017). These data appear to be at odds with the prominent E/I balance theory, which posits that AMPA EPSCs—more extensively studied than other channels such as NMDARs—and GABA IPSCs are balanced in neurons for optimal neuronal function (Turrigiano, 2012). As compared to AMPARs, however, the functional regulation of NMDAR-mediated transmission by pharmacological manipulation of GABA$_A$Rs has only been sparsely demonstrated (Pérez-Otaño and Ehlers, 2005; Watt et al, 2000). Furthermore, while it has been well-established that NMDAR activity can modulate GABA$_A$R trafficking and synaptic targeting and inhibitory transmission (Chiu et al, 2018; Gu et al, 2016; Horn and Nicoll, 2018; Wu et al, 2021), whether NMDAR-mediated transmission is regulated by GABA$_A$Rs at the single-cell level remains unknown.

## Results

### Functional GABA$_A$Rs require the β subunit in vivo

Although Cas9 has been extensively used in a range of experiments involving cell lines and embryos, applying Cas9 in vivo presents significant challenges (Platt et al, 2014). To explore the cell-autonomous role of GABA$_A$Rs in vivo, we utilized the CRISPR-Cas9 technology to develop a KO construct containing three single-guide RNA (sgRNA) sequences for β1, β2, and β3, respectively, chained together (Fig. 1A). Specifically, AAV2/9-β123gRNA-CaMKIIα-EGFP-Cre (hereafter β123-CRISPR) and AAV2/9-CaM-KIIα-EGFP-Cre (control) viral vectors were designed, and AAV virus was injected into bilateral CA1 regions of dorsal hippocampi of Cre-dependent Cas9 transgenic mice at P28-P35, hereafter referred to as β123-CRISPR mice and control mice, respectively (Fig. 1B,C). Around 4 weeks after injection, the mice were perfused and fixed, and their brain sections were processed for immunohistochemistry (Fig. 1D), which revealed abundant yet spatially restricted EGFP-Cre expression within the hippocampal CA1 region.

Electrophysiological recordings in EGFP-positive hippocampal CA1 neurons revealed an essential loss of mIPSCs (Fig. 1E) in neurons expressing β123-CRISPR compared to control, indicating that functional GABA$_A$Rs require inclusion of the β subunit in vivo. Consistently, in neurons expressing β123-CRISPR, we failed to detect spontaneous IPSCs (sIPSCs) (Fig. EV1). The lack of GABA$_A$R-mediated inhibitory currents at the level of individual neurons allowed us to investigate the cell-autonomous role of GABA$_A$Rs in the regulation of excitatory synaptic function.

### Single-cell elimination of GABA$_A$Rs impairs NMDAR-, but not AMPAR-, mediated synaptic transmission

Whole-cell recording of NMDAR- and AMPAR-mediated EPSCs in CA1 pyramidal neurons from β123-CRISPR mice revealed that NMDA/AMPA ratio was dramatically reduced in pyramidal neurons lacking GABA$_A$Rs (Fig. 2A), showing a deficit in excitatory synaptic transmission. To further examine cell-autonomous role of GABA$_A$Rs in the regulation of excitatory postsynaptic function, the input–output properties of NMDAR- and AMPAR-mediated EPSCs were separately measured. Notably, NMDAR-mediated

EPSCs in neurons expressing β123-CRISPR were significantly decreased by ~60%, while AMPAR-mediated EPSCs were slightly but not significantly decreased, compared to control neurons (Fig. 2B,C). Consistently, neither the amplitude nor the frequency of AMPAR-mediated mEPSCs showed a significant change in neurons expressing β123-CRISPR (Fig. EV2). Importantly, paired-pulse ratio (PPRs) of AMPA EPSCs, a measure of presynaptic neurotransmitter release probability, was not altered in β123-CRISPR neurons (Fig. 2D,E), showing that loss of GABAergic transmission does not change presynaptic neurotransmitter release probability at excitatory synapses.

To exclude potential contributions from network-level effects, we recorded NMDAR-mediated EPSCs from three groups of hippocampal CA1 neurons: GFP-positive neurons in control mice, and both GFP-positive and GFP-negative neurons in β123-CRISPR mice. While GFP-positive neurons expressing β123-CRISPR showed a significant reduction in NMDAR-EPSCs, GFP-negative neurons from the same regions did not differ significantly from control GFP-positive neurons (Fig. EV3). These results support the conclusion that the observed reduction in NMDAR transmission is specific to β123-CRISPR expressing neurons and not due to a generalized network effect.

Taken together, these data demonstrate that at single-cell level, genetic deletion of GABA$_A$Rs induces a specific reduction of NMDAR- but not AMPAR-mediated excitatory synaptic transmission.

### Rescue of NMDAR-mediated whole-cell currents depends on specific GABA$_A$R subtypes

It has been shown that GABA$_A$R β3 subunit plays a critical role in maintaining inhibitory transmission and is able to compensate for the loss of the other two β subunits in hippocampal neurons (Nguyen and Nicoll, 2018). In addition, it has been demonstrated that epilepsy-associated pathogenic *GABRB3* Tyr302Cys (Y302C) variants can lead to loss-of-function receptors, and patients with this variant have severe developmental and epileptic encephalo-pathies (Absalom et al, 2022; Shi et al, 2019). As Cas9 has been broadly applied in vitro (Duan et al, 2019; Nguyen and Nicoll, 2018; Platt et al, 2014), we performed genetic rescue experiments by developing gRNA-resistant β1, β2, β3 (β1*, β2*, or β3*, respectively) and β3*Y302C. These constructs were first validated in HEK293 cells (Duan et al, 2019; Fig. EV4A). To better reflect physiological conditions and confirm our findings in a native neuronal context, we subsequently conducted rescue experiments in primary hippocampal neuronal cultures. The gRNA-resistant constructs were co-transfected with β1–3 gRNAs in hippocampal neuronal cultures at DIV 3, followed by electrophysiological analysis at DIV 17-21 (Fig. 3A–D). Electrophysiological data showed that deletion of all three β1, 2, and 3 subunits in primary neuronal cultures essentially eliminated GABA-evoked whole-cell currents, which could be fully rescued by co-expressing β2* or β3*, and partially rescued by β1*, consistent with previous reports (Duan et al, 2019; Nguyen and Nicoll, 2018). Interestingly, loss of GABA-evoked currents in neurons expressing β1–3 gRNAs was not rescued by co-expressing β3*Y302C, showing that Y302C mutation disables β3 in rescuing GABA$_A$R-mediated currents (Fig. 3C). Correspondingly, loss of GABA$_A$Rs in neurons expressing β1–3 gRNAs induced a dramatic decrease in NMDA-evoked whole-cell

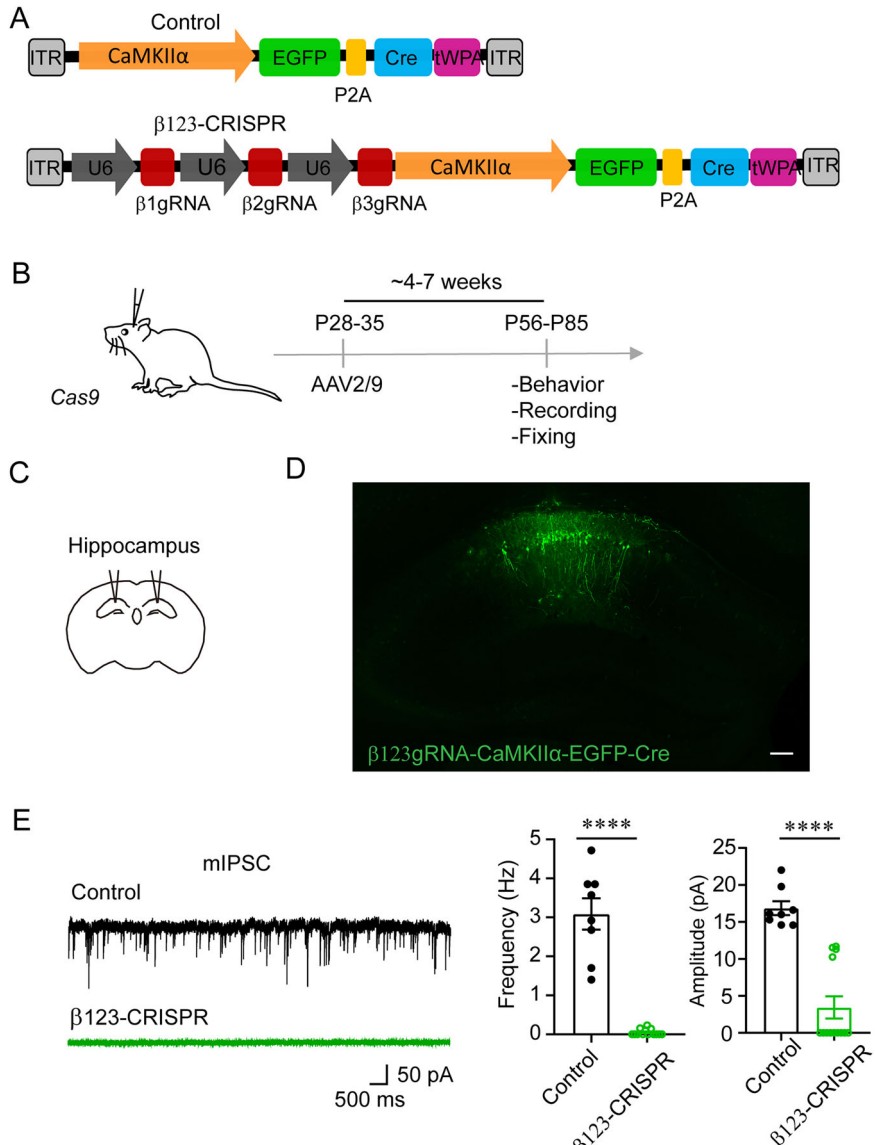

**Figure 1. Functional GABA$_A$Rs require the β subunit in vivo.**

(A) Schematic representation of KO constructs utilized for expressing the sgRNA targeting GABA$_A$R β1, β2, and β3 subunits. (B, C) Experimental outline illustrating the viral injection schedule targeting the hippocampi of Cre-dependent Cas9 transgenic mice. (D) Representative images show robust yet spatially restricted expression of EGFP-Cre in CA1 hippocampal neurons. Scale bars, 100 μm (left) and 25 μm (right). (E) Genetic deletion of GABA$_A$R β1, β2, and β3 subunits eliminated GABAergic synaptic transmission in hippocampal CA1 neurons expressing β123-CRISPR compared to control (bar graphs indicate mean ± SEM, $n = 8$, 13 from 3 or 4 mice, respectively; unpaired $t$ test, ****$P < 0.0001$). Source data are available online for this figure.

currents (Fig. 3D). Importantly, the reduction of NMDA-evoked whole-cell currents was rescued by co-expression of β2* or β3* subunits, the dominant β isoforms expressed in the brain, but not significantly rescued by β1* or β3*Y302C (Fig. 3D). These results suggest that sufficient expression of GABA$_A$Rs at the cell surface in individual neurons is crucial for functional expression of NMDARs at the neuronal surface.

To test the KO effect of GABA$_A$Rs on AMPARs in hippocampal neuron cultures, we performed the same experiment and found that compared with controls, AMPAR-mediated whole-cell currents was not significantly changed in neurons expressing β1–3 gRNAs (Fig. EV4B), further showing that genetic deletion of GABA$_A$Rs

selectively impairs NMDAR-, but not AMPAR-, mediated whole-cell currents in hippocampal neurons.

To examine surface expression of NMDARs in neurons lacking functional GABA$_A$Rs, we performed immunocytochemical experiments. We found that the surface expression of GluN1 subunits, which are required for functional formation of NMDARs (Nakazawa et al, 2004), was significantly reduced at dendrites in cultured hippocampal neurons expressing β123-gRNAs (Fig. EV4C).

Together, these data demonstrate that both NMDAR-mediated whole-cell currents and surface expression critically rely on the presence of functional GABA$_A$Rs.

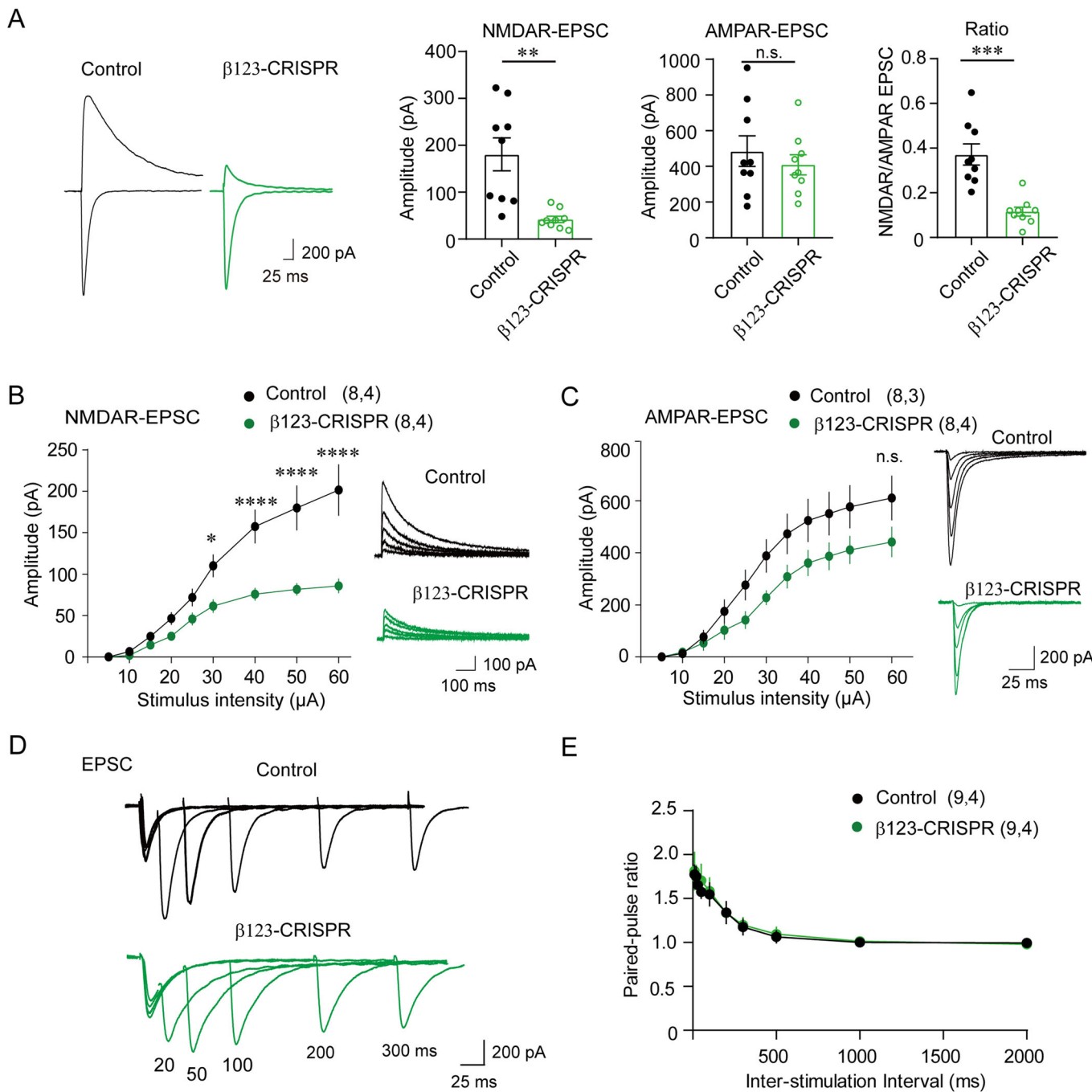

**Figure 2. Single-cell elimination of GABA$_A$Rs impairs NMDAR-, but not AMPAR-, mediated synaptic transmission in hippocampal CA1 neurons.**

(A) NMDAR-EPSC to AMPAR-EPSC ratio was dramatically reduced in CA1 pyramidal neurons expressing β123-CRISPR compared to control vector. Bar graphs indicate mean ± SEM, relative NMDAR-EPSC, **$P = 0.0013$; relative AMPAR-EPSC, n.s., not significant, $P = 0.7108$; NMDA to AMPA EPSC ratios, ***$P = 0.0001$, $n = 9$, $N = 4$, unpaired $t$ test. (B) In the presence of the AMPAR antagonist CNQX, hippocampal CA1 neurons expressing β123-CRISPR exhibited a reduced input–output curve of NMDAR-EPSCs compared to control neurons. Line graphs indicate mean ± SEM, *$P = 0.0199$, ****$P < 0.0001$, $n = 8$, $N = 4$. Data were analyzed using two-way repeated-measures ANOVA (factors: group × stimulation intensity), followed by Sidak's multiple comparisons test at each intensity. (C) In the presence of the NMDAR antagonist APV, β123-CRISPR neurons showed a modest but statistically non-significant change in the input–output curve of AMPAR-EPSCs. Line graphs indicate mean ± SEM, $P = 0.26$, $n = 8$, $N = 3$ or 4. Data were analyzed using two-way repeated-measures ANOVA followed by Sidak's multiple comparisons test at each intensity. (D, E) Paired-pulse ratio (PPRs) of AMPA EPSCs was not altered in β123-CRISPR neurons (Line graphs indicate mean ± SEM, $P > 0.9999$, $n = 9$, $N = 4$, two-way repeated-measures ANOVA, factors: group × stimulation interval). Source data are available online for this figure.

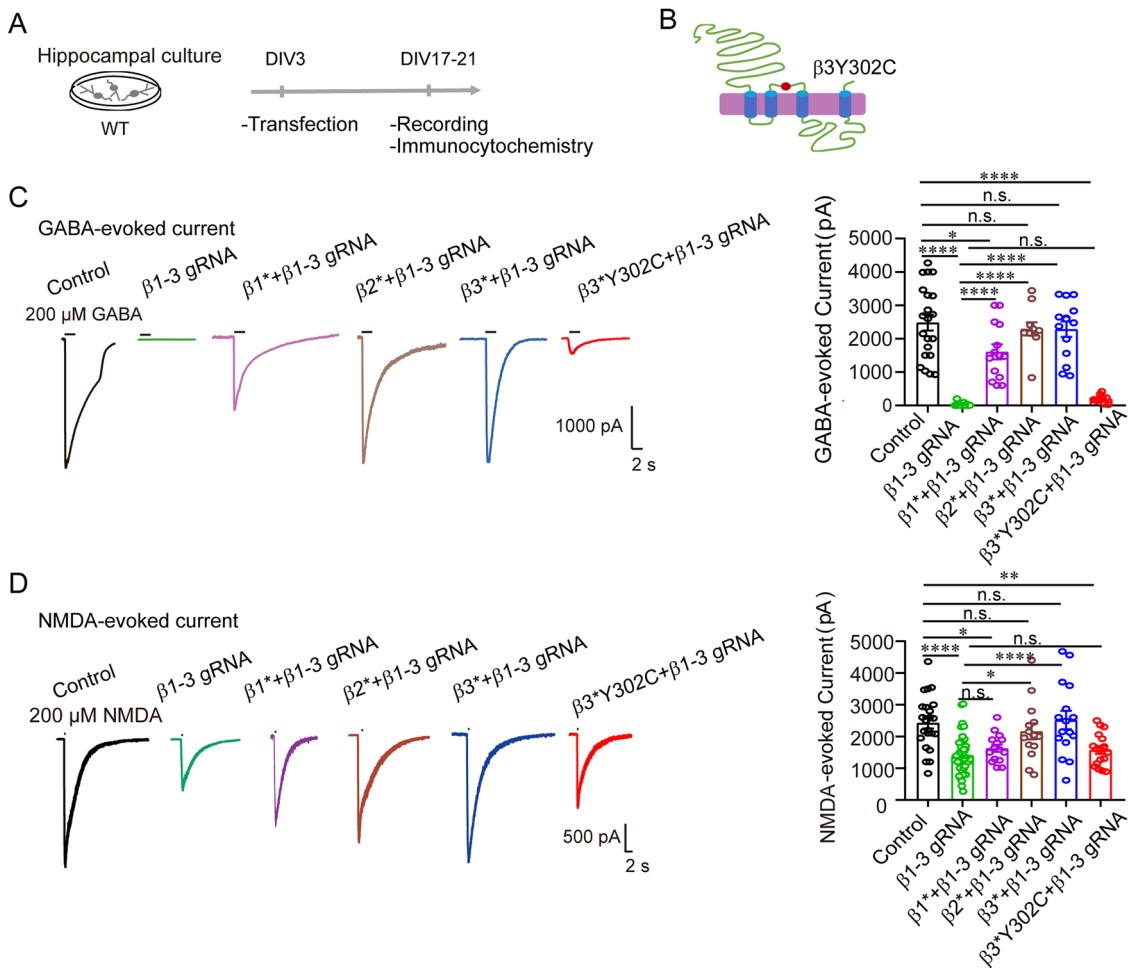

**Figure 3. Genetic rescue of NMDAR-mediated whole-cell currents depends on specific GABA$_A$R subtypes in hippocampal cultures.**

(A) Experimental design for electrophysiological recording and immunocytochemistry. (B) Schematic of point mutation in β3Y302C (indicated by the red dot) used for electrophysiological experiments. (C) Genetic deletion of β1, 2, and 3 subunits in primary hippocampal neuronal cultures resulted in the essential elimination of GABA-evoked whole-cell currents, which could be rescued by co-expressing β2* or β3*. Partial rescue was observed with co-expression of β1*. However, co-expression with β3*Y302C did not result in rescue. Control, $n = 21$; β1–3 gRNA, $n = 16$, β1* + β1–3 gRNA, $n = 14$; β2* + β1–3 gRNA, $n = 11$; β3* + β1–3gRNA, $n = 14$; β3*302C + β1–3 gRNA, $n = 11$, $N = 3$; Bar graphs indicate mean ± SEM, ****$P < 0.0001$, *$P = 0.0208$, n.s., $P > 0.9999$, One-way ANOVA followed by the Bonferroni test. (D) Loss of GABA$_A$Rs in neurons expressing β1–3 gRNAs induced a dramatic decrease in NMDA-evoked whole-cell currents. The reduction of NMDA-evoked whole-cell currents were rescued by co-expression of β2* or β3* subunits, but not significantly rescued by β1* or β3*Y302C. Control, $n = 23$; β1–3 gRNA, $n = 42$; β1* + β1–3 gRNA, $n = 15$; β2* + β1–3 gRNA, $n = 14$; β3* + β1–3gRNA, $n = 16$; β3*302C + β1–3 gRNA, $n = 17$; $N = 6$; Bar graphs indicate mean ± SEM, ****$P < 0.0001$, *$P = 0.0278$, n.s. $P = 0.9949$, n.s. $P > 0.9999$, **$P = 0.0091$; n.s. $P = 0.9981$, *$P = 0.0261$, ****$P < 0.0001$, n.s., $P = 0.9999$, one-way ANOVA followed by the Bonferroni test. Bars indicate GABA, NMDA applications for GABA-evoked, or NMDA-evoked whole-cell currents, respectively. Source data are available online for this figure.

## Neuronal excitability governs NMDAR-mediated synaptic transmission in neurons lacking GABA$_A$Rs

Activity-dependent NMDAR targeting is essential for synapse formation and plasticity (Mu et al, 2003). To explore how activity-dependent mechanisms influence NMDAR synaptic function in single neurons, we conducted electrophysiological recordings in hippocampal CA1 neurons from β123-CRISPR mice to assess the cell-autonomous effects of GABA$_A$R loss on neuronal excitability. Our results revealed that GABA$_A$R loss significantly increased evoked action potential firing in hippocampal CA1 neurons in brain slices (Fig. EV5A,B), with similar findings observed in hippocampal neuronal cultures (Fig. 4A). Additionally, β123-CRISPR mice exhibited heightened seizure severity following PTZ

administration (40 mg/kg) (Fig. EV5C). These findings demonstrate that the loss of GABA$_A$Rs enhances neuronal excitability, increasing seizure susceptibility.

To test whether homeostatic reduction of NMDARs in neurons lacking GABA$_A$Rs is related to elevated activity, we genetically suppressed neuronal excitability by expressing Kir2.1, using Kir2.1Mut as a non-conducting control at the single-cell level (Lin et al, 2010; Xue et al, 2014). These constructs were transfected alone or co-transfected with β1–3 gRNAs in hippocampal neuronal cultures at DIV 3-4, followed by electrophysiological analysis at DIV13-16. Current-clamp recordings showed that neurons expressing Kir2.1 alone had a marked reduction in evoked action potential firing compared to controls, while expression of the mutant form, Kir2.1Mut, had no such effect (Fig. 4). Notably, the

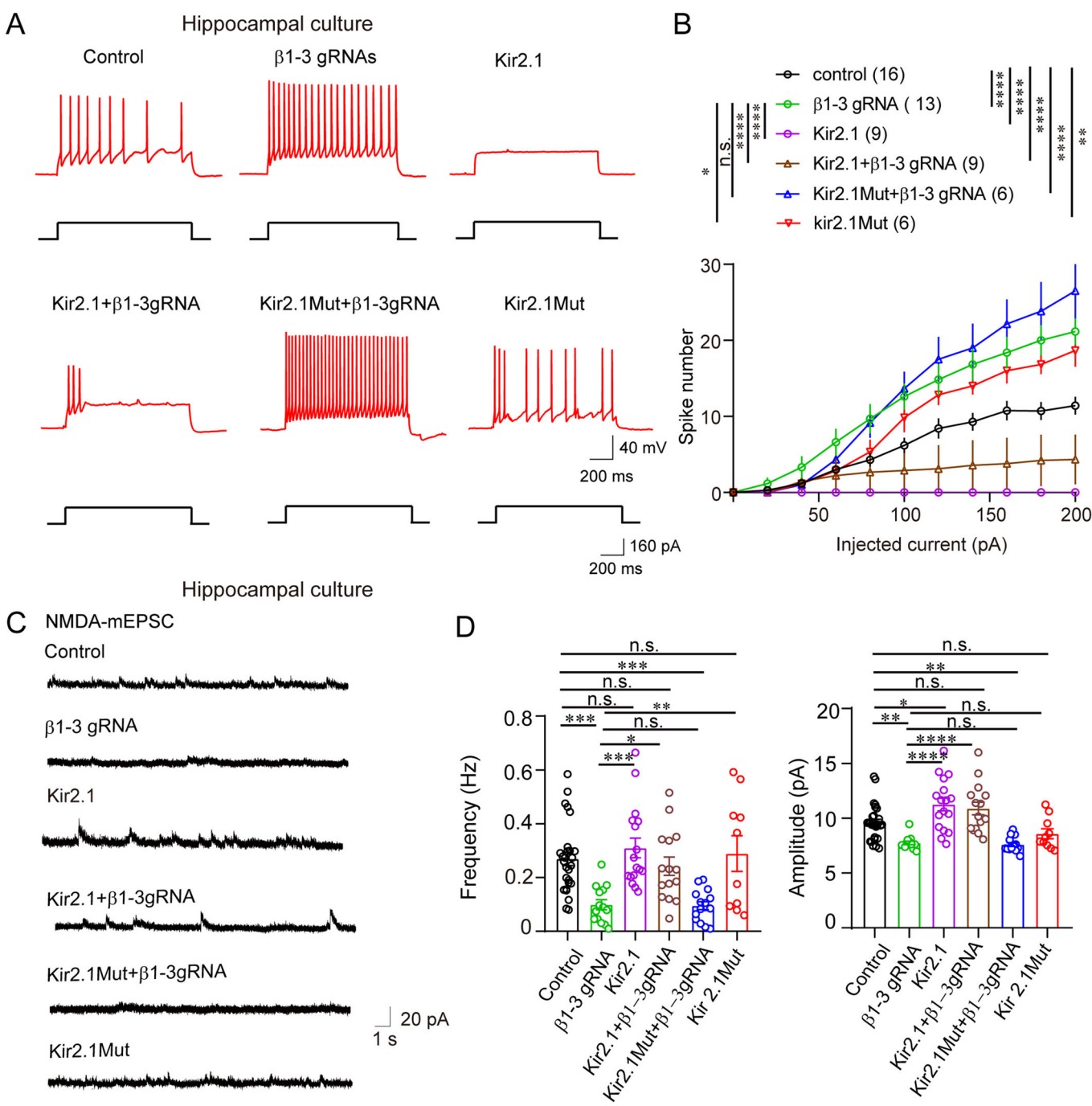

**Figure 4. Neuronal excitability governs NMDAR-mediated synaptic transmission in neurons lacking GABA_ARs.**

(A) The increased frequency of evoked action potential firing observed with loss of GABA_ARs in neurons was significantly reduced by co-expression of Kir2.1, but not by non-conducting Kir2.1Mut in hippocampal neuronal cultures. (B) Quantification of neuronal excitability across experimental groups: control, $n = 16$; β1–3 gRNA, $n = 13$; Kir2.1, $n = 9$; Kir2.1 + β1–3 gRNA, $n = 9$; Kir2.1 Mut+β1–3 gRNA, $n = 6$; Kir2.1 Mut, $n = 6$. Graph shows the number of action potentials evoked in response to incremental current injections. Line graphs indicate mean ± SEM, $*P = 0.0217$; $**P = 0.03$; $****P < 0.0001$; n.s. $P = 0.6936$. Statistical analysis was performed using two-way ANOVA, followed by Sidak's multiple comparisons test. (C) Example recordings of NMDA-mEPSC from neurons transfected with control vector, β1–3 gRNA, Kir2.1, Kir2.1 + β1–3 gRNA, Kir2.1 Mut+β1–3 gRNA, or Kir2.1 Mut in hippocampal neuronal cultures. (D) Bar graphs illustrate that the loss of GABA_ARs significantly reduced the frequency and amplitude of NMDA-mEPSC in neurons lacking GABA_ARs. Reducing excitability by co-expressing Kir2.1 effectively restored NMDA-mEPSC frequency and amplitude, whereas the co-expression of a non-conducting Kir2.1Mut had no effect. control, $n = 27$; β1–3 gRNA, $n = 14$; Kir2.1, $n = 17$; Kir2.1 + β1–3 gRNA, $n = 16$, Kir2.1 Mut+β1–3 gRNA, $n = 15$; and Kir2.1 Mut, $n = 10$. Bar graphs indicate mean ± SEM, $**P = 0.0019$, n.s. $P = 0.9115$, n.s. $P = 0.9870$, $***P = 0.001$, n.s. $P = 0.9985$; $***P = 0.0003$; $*P = 0.043$; n.s. $P > 0.9999$, $**P = 0.0093$. One-way ANOVA with Bonferroni's test. Source data are available online for this figure.

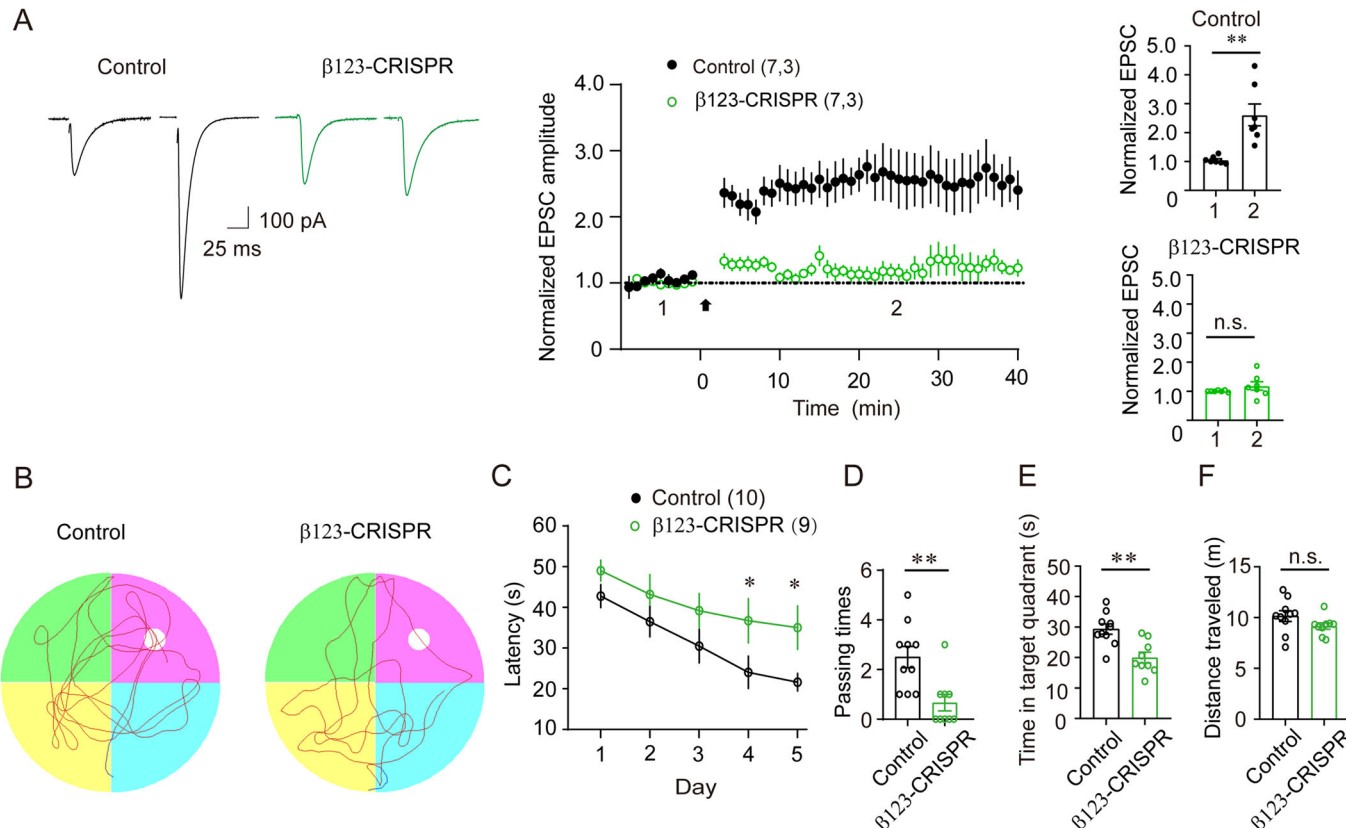

**Figure 5.  Loss of GABA_ARs in hippocampal CA1 pyramidal neurons abolishes NMDAR-dependent LTP and impairs spatial memory.**

(A) Time-course changes of EPSC amplitude (normalized to baseline) following a pairing protocol (0 mV, 2 Hz, 360 pulses; arrow), measured from hippocampal CA1 neurons in either control or β123-CRISPR mice. Representative traces before and at 20–30 min after pairing were shown on the left. The bar graph on the right indicates that LTP was abolished in β123-CRISPR mice, whereas it remained normal in control mice. "1" denotes the baseline, while "2" represents the measurements taken during LTP assessment at 20–30 min after pairing. Bar graphs indicate mean ± SEM, **$P = 0.0015$, n.s, not significant, $P = 0.3739$, $n = 7$, $N = 3$, unpaired $t$-test. (B) Representative probe test trajectories of the swimming tracks from mice expressing control vector or β123-CRISPR in hippocampal CA1 region during the Morris water maze test. (C) Latency to the platform of control or β123-CRISPR mice from days 1 to 5 (Day 4, *$P = 0.04$; Day 5, *$P = 0.02$, $n = 10, 9$, unpaired $t$ test). Mice were given four trials per day and data represent the mean ± SEM of blocks of four trials. Data are latency(s) to reach the goal, using identical sequences of start locations. (D) Platform frequency of mice in the probe trial at day 6 (Bar graphs indicate mean ± SEM, **$P = 0.004$, $n = 10, 9$, unpaired $t$ test). (E) Time spent in the target quadrant during the probe trial (bar graphs indicate mean ± SEM, **$P = 0.0013$, $n = 10, 9$, unpaired $t$ test). (F) The average swim distance of control and β123-CRISPR mice during the probe test (Bar graphs indicate mean ± SEM, $P = 0.138$, $n = 10, 9$, unpaired $t$ test). The small tank was 120 cm in diameter. All testing conditions and trials were identical for the two groups. Source data are available online for this figure.

increased frequency of evoked action potential firing observed with loss of GABA_ARs in neurons was significantly reduced by co-expression of Kir2.1, but not by Kir2.1Mut (Fig. 4A,B). These results suggest that reducing neuronal excitability via Kir2.1 can counteract the hyperexcitability caused by GABA_AR loss, while the non-conducting Kir2.1Mut lacks this capacity.

Voltage-clamp recordings revealed that loss of GABA_AR in neurons expressing β1–3 gRNA, which increased neuronal excitability, resulted in a significant reduction in the frequency and amplitude of NMDA-mEPSCs (Fig. 4C,D). Notably, co-expression of Kir2.1, which reduces excitability, effectively restored NMDA-mEPSC frequency and amplitude in neurons lacking GABA_ARs. In contrast, co-expression of the non-conducting Kir2.1Mut had no significantly effects (Fig. 4D). These findings illustrate that neuronal excitability governs NMDAR-mediated synaptic transmission in neurons lacking GABA_ARs. This underscores the pivotal

role of neuronal excitability in modulating synaptic NMDAR function in a cell-autonomous manner.

## Loss of GABA_ARs in hippocampal CA1 pyramidal neurons abolishes LTP and impairs spatial memory

LTP has been proposed to be a physiological substrate for long-term memory, and its induction requires activation of NMDARs in hippocampal neurons (Bliss and Collingridge, 1993). To examine whether NMDAR deficiency in neurons lacking GABA_ARs led to changes of excitatory synaptic plasticity, we employed whole-cell recordings to measure LTP at Schaffer collateral-CA1 synapses in CA1 neurons. We found that in control neurons LTP was readily induced by a pairing protocol (0 mV, 2 Hz, 360 pulses, Fig. 5A). In contrast, LTP was abolished in neurons lacking GABA_ARs (Fig. 5A).

The Morris water maze (MWM) is a robust test for hippocampal function due to its strong correlation with synaptic plasticity and NMDAR function, and studies show that hippocampal lesions or NMDA deficiency within the hippocampus consistently impair performance in this task (Nakazawa et al, 2004; Vorhees and Williams, 2006). To study whether the loss of LTP in dorsal hippocampal CA1 neurons lacking GABA$_A$Rs affected spatial memory, adult male Cas9 mice expressing β123-CRISPR or control vector specifically in the dorsal hippocampal CA1 pyramidal neurons were used for the MWM test. We found that control mice exhibited a significantly greater reduction in escape latency from block 1 to block 5. In contrast, β123-CRISPR mice exhibited a slight but statistically insignificant progressive improvement in escape behavior, they consistently showed a longer escape latency over training blocks as compared to control mice (Fig. 5C). Notably, β123-CRISPR mice had significantly higher escape latencies than control mice on blocks 4 and 5 (Fig. 5C). Moreover, in the subsequent probe trial, β123-CRISPR mice showed a significantly lower frequency of platform crossings and spent less time in the platform zone compared to controls (Fig. 5B,D,E). Notably, swim distance remained comparable between groups, effectively ruling out motor deficits as a confounding factor (Fig. 5F). Together, these data demonstrate that GABA$_A$Rs in hippocampal CA1 pyramidal neurons are critical both for LTP induction and spatial memory.

Reduced synaptic NMDAR function may contribute to impairment of spatial memory in β123-CRISPR mice. To test this hypothesis and restore NMDAR function, we administered D-cycloserine (DCS), an NMDAR partial agonist (Kochlamazashvili et al, 2012; Walker et al, 2002), to β123-CRISPR mice (Fig. 6A). Although DCS did not affect LTP in CA1 pyramidal neurons from hippocampal slices prepared from 3-month-old control mice (Kochlamazashvili et al, 2012), acute application of DCS restored LTP in β123-CRISPR neurons to the levels similarly to that observed in control mice (Fig. 6A). The Y maze spontaneous alternation assay requires a functionally intact dorsal hippocampus and has also been widely used to test spatial working memory (Lippi et al, 2016). Given its sensitivity to hippocampal dysfunction and compatibility with the short-term pharmacological effects of DCS, we selected the Y-maze as an alternative approach to evaluate hippocampal function in CRISPR β1–3 mice. In the Y maze test, we found that though genetic deletion of GABA$_A$Rs in hippocampal CA1 neurons in β123-CRISPR mice led to spatial memory deficits (Fig. 6B,C), injection of DCS rapidly restored the Y maze spontaneous alternation performance while the mice were traveling the similar distance during the test (Fig. 6B–D). These results indicate that reduced NMDAR-mediated synaptic transmission may contribute to the impairment of LTP and spatial memory in β123-CRISPR mice.

## Discussion

The cell-autonomous role of GABA$_A$Rs in the regulation of excitatory synaptic function has not been examined in depth, and it remains unclear whether AMPARs and NMDARs are similarly regulated by GABA$_A$Rs at the single cell level. Using a single-cell genetic approach, we have identified that in hippocampal neurons lacking GABA$_A$Rs, NMDAR-, but not AMPAR-, mediated transmission is strongly reduced. Consequently, NMDAR-

dependent LTP and spatial memory are impaired. These findings uncover a novel excitation-inhibition balance driven by the interaction between NMDARs and GABA$_A$Rs, advancing our understanding of synaptic plasticity and its potential relevance to brain disorders characterized by disrupted GABAergic and NMDAR signaling.

## GABA$_A$Rs are required for NMDAR-mediated synaptic transmission

In this study, by targeting the β1–3 subunits, we have demonstrated that the native assembly of functional GABA$_A$R in vivo requires the incorporation of the β subunits. The efficacy of the GABA$_A$R KO vector was previously validated through immunostaining and electrophysiology in primary cultured neurons, which confirmed the loss of β1, β2, and β3 subunit expression and the absence of mIPSCs (Duan et al, 2019). In the present study, we further show that deletion of these β subunits abolished GABA-evoked whole-cell currents. In vivo, the elimination of GABA$_A$Rs in hippocampal CA1 pyramidal neurons led to the silencing of GABAergic transmission, underscoring the specificity and effectiveness of the KO strategy.

Our data demonstrate that at the level of individual neurons, the loss of functional GABA$_A$Rs does not significantly change AMPAR-mediated excitatory transmission or whole-cell currents, but selectively decreases the NMDA/AMPA current ratio, NMDAR-mediated excitatory currents or whole-cell response, accompanied by the loss of NMDAR-dependent LTP and impairment of spatial memory. Previous studies have shown that glutamate primarily acting via NMDARs controls GABAergic synapse development and GABA$_A$R-mediated synaptic transmission (Chiu et al, 2018; Gu et al, 2016; Horn and Nicoll, 2018; Wu et al, 2021). Our data have now indicated that GABA acting via GABA$_A$Rs can selectively modulate NMDAR, but not AMPAR, function at the single cell level. The reciprocal regulation between GABA$_A$Rs and synaptic NMDAR function at the level of individual cells may provide a fine-tuning mechanism to balance inhibitory and excitatory synaptic strength.

Although mEPSC frequency is typically used to assess presynaptic function, postsynaptic factors can also influence it (Scheefhals and MacGillavry, 2018). Given that the PPRs of AMPA EPSCs was not altered in β123-CRISPR neurons, but the NMDA-evoked whole-cell currents and the surface staining of GluN1 receptors were dramatically reduced, we propose that GABA$_A$Rs in hippocampal neurons are crucial for postsynaptic NMDARs function, leading to a reduction of NMDAR-mEPSC frequency. The depression of NMDAR responses occurs independently of AMPAR response changes further supporting a postsynaptic mechanism.

Our genetic rescue experiments demonstrate that overexpression of the β2* or β3* subunit, but not the β1* subunit or the loss-of-function mutant β3*Y302C, in individual GABA$_A$Rs-lacking neurons successfully rescues NMDA-evoked whole-cell currents. This highlights the crucial and functionally sufficient role of GABA$_A$Rs in regulating the functional surface expression of NMDARs. Additionally, our findings reveal that the rescue of NMDAR functional expression at the neuronal surface depends on GABA$_A$R β2 and β3, but not β1 subunits. Although the use of gRNA-resistant GABA$_A$Rs subunits may generate receptors with

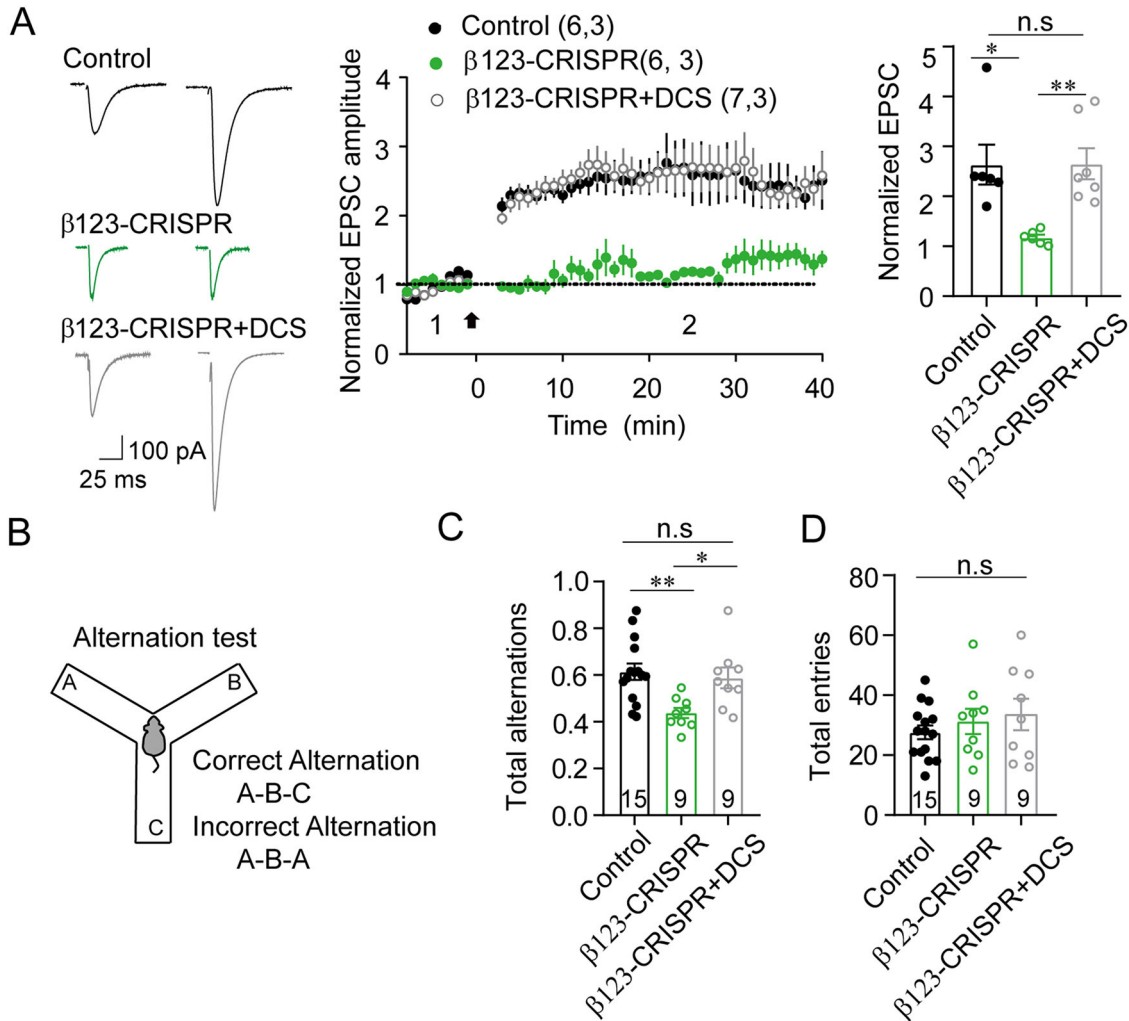

**Figure 6. Pharmacological restoration of NMDAR function rescued both LTP and spatial memory deficits in β123-CRISPR mice.**

(A) Time-course changes of EPSC amplitude (normalized to baseline) following a pairing protocol (0 mV, 2 Hz, 360 pulses; arrow) measured from control and β123-CRISPR mice (with or without application of ᴅ-cycloserine, DCS). Representative traces before and at 20–30 min after pairing are shown on the left. "1" denotes the baseline, while "2" represents the measurements taken during LTP assessment at 20–30 min after pairing. The bar graph on the right showed that administration of DCS restored LTP in β123-CRISPR mice. Control vs β123-CRISPR, *$P = 0.0117$; Control vs β123-CRISPR + DCS, n.s., not significant, $P > 0.99$; β123-CRISPR vs β123-CRISPR + DCS, **$P = 0.0083$; Line and bar graphs indicate mean ± SEM, Control, $n = 6$; β123-CRISPR, $n = 6$; β123-CRISPR + DCS, $n = 7$; $N = 3$, one-way AVOVA with Bonferroni's test. (B) Alternation test involving a symmetrical Y maze with examples of correct and incorrect alternations. (C) Genetic deletion of GABA$_A$Rs in hippocampal CA1 neurons in β123-CRISPR mice led to spatial memory deficits, while injection of DCS rapidly restored the Y maze spontaneous alternation performance (bar graphs indicate mean ± SEM, Control $n = 15$, β123-CRISPR $n = 9$, β123-CRISPR + DCS $n = 9$, **$P = 0.0046$, n.s. $P = 0.9420$, *$P = 0.0366$, one-way AVOVA with Bonferroni's test). (D) The mice were traveling the similar distance during the test (Bar graphs indicate mean ± SEM, Control $n = 15$, β123-CRISPR $n = 9$, β123-CRISPR + DCS $n = 9$, n.s., not significant, $P = 0.4931$, one-way AVOVA with Bonferroni's test). Source data are available online for this figure.

non-native compositions and potentially altered kinetics, such differences do not affect the main conclusions of this study. Future investigations will need to explore mechanisms by which GABA$_A$R β subunit contribute to the regulation of the functional surface expression of NMDARs, and to identify the specific GABA$_A$R subunits that are functionally restored during this process.

While the dynamic regulation of postsynaptic NMDARs is recognized as crucial for synapse maturation and plasticity (Nakazawa et al, 2004), most research to date has primarily focused on the homeostatic regulation of AMPARs (Turrigiano, 2012), leaving the mechanisms underlying NMDAR homeostasis poorly understood. Our findings that genetic elimination of

GABA$_A$Rs at single cell levels significantly increased neuronal excitability, leading to reduced NMDA-mEPSC frequency and amplitude, which were restored by reducing excitability through Kir2.1 co-expression, demonstrate that neuronal excitability governs synaptic NMDAR function in neurons lacking GABA$_A$Rs. These results highlight a cell-autonomous mechanism by which neuronal activity modulates synaptic NMDARs to counteract pathological hyperactivity, providing new molecular mechanistic insights into the E/I balance between NMDARs and GABA$_A$Rs.

Previous studies, along with our current findings, demonstrate that regulation of NMDARs can occur independently (Goold and Nicoll, 2010). Indeed, it has been shown that signaling pathways

regulating synaptic removal of NMDARs and AMPARs diverge (Goold and Nicoll, 2010). The presence of high-mannose glycans on NMDAR-GluN1 subunits makes them highly susceptible to ubiquitination and endoplasmic reticulum-associated degradation (ERAD), enabling the activity-dependent ubiquitin-proteasome pathway to mediate the synaptic removal of NMDARs, thereby affecting their expression levels and currents (Ehlers, 2003; Kato et al, 2005; Scott and Panin, 2014). However, N-glycosylation is not universally required for AMPAR function (Scott and Panin, 2014), and the activity-dependent ubiquitin-proteasome pathway appears to have a differential role in regulating AMPARs (Widagdo et al, 2017). While GluA1 ubiquitination was suggested to signal AMPAR endocytosis (Widagdo et al, 2017), other studies disputed this, showing that neither GluA1 nor GluA2 ubiquitination regulates AMPAR surface expression or agonist-induced endocytosis (Lussier et al, 2011; Widagdo et al, 2015). Based on these findings, we propose a hypothesis in which the absence of GABA_ARs leads to elevated neuronal excitability, which in turn upregulates ERAD activity. This results in the degradation of improperly folded or glycosylated GluN1 subunits, thereby reducing NMDAR surface expression and function. This speculation provides a plausible explanation for the observed NMDAR deficits in the absence of GABA_ARs and suggests a potential mechanism through which GABA_ARs differentially influence synaptic NMDARs and AMPARs. Further investigation is needed to determine whether this mechanism accounts for the selective relationship observed between GABA_ARs and NMDARs.

For the last two decades, the E/I balance has been proposed as a foundational framework for investigating mechanisms in brain disorders (Sohal and Rubenstein, 2019). Here, we have discovered a new molecular mechanism that contributes to E/I balance, which is not based on the classic balance between synaptic AMPARs and GABA_ARs, but between synaptic NMDARs and GABA_ARs. This regulatory interaction plays a critical role in brain function and disorders. In Alzheimer's disease (AD), disruptions in inhibitory circuits, as well as the impairment of GABA_ARs, are notably observed (Chen et al, 2017; Ulrich, 2015; Varela et al, 2019). Likewise, there's a notable expression decrease in NMDAR GluN1 and GluN2B subunits in AD brains (Mishizen-Eberz et al, 2004), accompanied by the reduced NMDAR function (Lambert et al, 1998; Snyder et al, 2005). Similarly, in schizophrenia, both NMDAR hypofunction and deficits in GABA_ARs are strongly implicated. Hypofunction of selective GABA_AR populations causes schizophrenia-related cognitive impairment, and increasing GABA_AR function or trafficking pharmacologically may alleviate some of the symptoms (Cohen et al, 2015). Studies have also found that NMDAR antagonists can mimic primary schizophrenia symptoms in healthy humans and exacerbate symptoms in patients with schizophrenia (Paoletti et al, 2013). Thus, an E/I balance involving NMDAR-GABA_AR interplay, in addition to the traditional AMPAR-GABA_AR interaction, may carry substantial significance in understanding cognitive disorders like AD and schizophrenia.

## GABA_ARs are crucial for LTP and hippocampus-dependent learning and memory

Previous studies on hippocampal GABAergic signaling have shown mixed effects on LTP and memory, with evidence suggesting that GABA_AR inhibition can either enhance or impair these processes (Bast et al, 2017; Casasola et al, 2004; Ferando et al, 2016; Lister, 1985; Wigstrom and Gustafsson, 1983). The apparent differences in these previous conflicting conclusions are probably due to enormous regional diversity and complexity of GABA_ARs, complicated neural circuit effects, and limited selectivity of GABA_AR pharmacological tools (Burrone et al, 2002).

With genetic manipulation of GABA_ARs targeting single hippocampal CA1 neurons, here we have found that single-cell silencing of GABAergic transmission led to a significant reduction in NMDAR-mediated synaptic transmission, loss of LTP, and impairment in spatial memory. In addition to the direct effect of GABA_ARs in regulating learning and memory (Engin et al, 2018), our research demonstrates that their modulation of NMDAR-mediated synaptic transmission is critical for LTP and spatial memory. Previous studies have shown that DCS can fully restore NMDAR function and improve social interaction in NMDAR hypofunction models, such as *Shank2 or Dock4* KO mice (Guo et al, 2021; Won et al, 2012). At the same dose, our findings indicate that DCS can rescue LTP in β123-CRISPR neurons and improve Y maze performance in β123-CRISPR mice. Our results indicate that reduced NMDAR-mediated synaptic transmission may contribute to impairment of spatial memory in β123-CRISPR mice. These results deepen our understanding of synaptic plasticity and highlight its implications for disorders characterized by disrupted GABAergic and NMDAR signaling. A limitation of our study is that because of the sparse expression of EGFP-Cre for GABA_AR deletion in hippocampal CA1 neurons, i.p. injection of DCS not only enhances NMDAR activity in hippocampal neurons lacking GABA_ARs, but also activates NMDARs in other intact neurons, glia or other cell types containing GABA_ARs (Verkhratsky and Kirchhoff, 2007), which may also contribute to behavioral phenotypes observed in this study. Future experiments with more precise manipulation of synaptic NMDAR function in hippocampal neurons lacking GABA_ARs will further the understanding of the cell-autonomous role of GABA_ARs in learning and memory.

In summary, through a combination of molecular, genetic, behavioral, pharmacological and electrophysiological approaches, our data have provided evidence that the cell-autonomous GABA_AR function is crucial for NMDAR-mediated synaptic transmission, plasticity and spatial memory. These data broaden our understanding the function of GABA_ARs at the single cell level, highlighting its significant influence on glutamatergic synaptic activity beyond the primary source of neural inhibition. Furthermore, we have also discovered a distinct type of E/I balance at the single-cell level, which is not based on the balance between synaptic AMPARs and GABA_ARs, but is dependent on synaptic NMDARs and GABA_ARs. Given that coordinated dysregulations of GABA_AR- and NMDAR-mediated synaptic function are evident in several cognitive disorders (Chen et al, 2017; Cohen et al, 2015; Lambert et al, 1998; Mishizen-Eberz et al, 2004; Paoletti et al, 2013; Snyder et al, 2005; Ulrich, 2015; Varela et al, 2019), our findings also offer insight into crosstalk between glutamatergic and GABAergic systems, as well as how dysregulation of this crosstalk could be involved in cognitive impairment conditions.

# Methods

### Reagents and tools table

| Reagent or resource | Source | Identifier |
| --- | --- | --- |
| **Antibodies** | | |
| Polyclonal rabbit antibody GluN1 | SYSY | Cat#: 114003 RRID:AB_2247722 |
| Anti-GABA(A) β1 Receptor | SYSY | Cat# 224 703 RRID:AB_2619939 |
| Anti-GABA(A) β 2 Receptor | Sigma-Aldrich | Cat# AB5561 RRID:AB_177524 |
| Anti-GABA(A) β 3 Receptor | Sigma-Aldrich | Cat# SAB2100880, RRID:AB_10605792 |
| **Bacterial and viral strains** | | |
| AAV2/9-β123gRNA-CaMKIIα-EGFP-Cre | OBiO Technology | N/A |
| AAV2/9-CaMKIIα-EGFP-Cre | OBiO Technology | N/A |
| **Chemicals, peptides, and recombinant proteins** | | |
| Papain | Worthington | Cat#: LK003176 |
| Trypsin inhibitor | Sigma-Aldrich | Cat#: T9253 |
| Neurobasal Medium | GIBCO | Cat#: 21103-049 |
| B27 Supplement | GIBCO | Cat#: 17504-044 |
| GlutaMax | GIBCO | Cat#: 35050-061 |
| DMEM medium | GIBCO | Cat#: 10569-010 |
| FBS | GIBCO | Cat#: 10437-028 |
| Penicillin Streptomycin | GIBCO | Cat#: 15140-122 |
| Poly-D-lysine | Sigma-Aldrich | Cat#: P6407 |
| APV | Abcam | Cat#: ab120003 |
| CNQX | TOCRIS | Cat#: 0190 |
| Tetrodotoxin | BIOTIUM | Cat#: 00060 |
| Picrotoxin | TOCRIS | Cat# 1128 |
| D-Cycloserine | Selleck | Cat# S1998 |
| γ-Aminobutyric acid | Sigma-Aldrich | Cat# A2129 |
| N-Methyl-D-aspartic acid | Sigma-Aldrich | Cat# M3262 |
| S-AMPA | Abcam | Cat#144483 |
| **Experimental models: cell lines** | | |
| HEK293T | ATCC | Cat# CRL-1126 |
| **Experimental models: organisms/strains** | | |
| Mouse: C57BL/6JGpt KI(Cas9) | Gem Pharma tech | Cat# T002249 |
| Mouse: wild-type C57L/6J mice | Beijing Vital River | N/A |
| **Oligonucleotides** | | |
| sgRNA targeting sequence: mouse GABRB1: GCCGCGAGGGCTTCGGGCGT | This paper | N/A |
| sgRNA targeting sequence: mouse GABRB2:CAGACAGCGGCGATTATTAA | This paper | N/A |
| sgRNA targeting sequence: mouse GABRB3: ACGGTCGACAAGCTGTTGAA | This paper | N/A |
| **Recombinant DNA** | | |
| pCAGGS-IRES-mCherry | Dr Roger Nicoll (UCSF) | N/A |
| pSpCas9(BB)-2A-GFP (PX458) | Addgene | Cat# 48138 |
| sgRNA-resistant GABA(A) β1 | This paper | N/A |

| Reagent or resource | Source | Identifier |
| --- | --- | --- |
| sgRNA-resistant GABA(A) β2 | This paper | N/A |
| sgRNA-resistant GABA(A) β3 | This paper | N/A |
| sgRNA-resistant GABA(A) β3 302 C | This paper | N/A |
| pCAG-Kir2.1-T2A-tdTomato | Dr Massimo Scanziani (UCSF) | Cat #60598 |
| pCAG-Kir2.1Mut-T2A-tdTomato | Dr Massimo Scanziani (UCSF) | Cat #60644 |
| **Software and algorithms** | | |
| ImageJ | NIH | https://imagej.nih.gov/ij/ |
| GraphPad Prism 7.0 | GraphPad | https://www.graphpad.com |
| Igor Pro | Wavemetrics | https://www.wavemetrics.com |
| TopscanTM3.0 | Clever Sys., Inc | https://cleversysinc.com/CleverSysInc/ |

## Animals

All animal experimental procedures were approved by the Animal Care and Use Committees of Sun Yat-sen University. Time-pregnant C57BL/6J at E18 were used for dissociated hippocampal neuron cultures. Male Cre-dependent Rosa26-LSL-Cas9-tdTomato mice (P28-30) were purchased from Gem Pharma Tech (Nanjing, China), housed and bred under a 12-h circadian cycle. Genotyping Cas9 mice was performed by PCR of genomic tail DNA using the primers:

common forward primer (5' to 3') CGGCATGGACGAGCTGTACAAG;

common reverse primer (5' to 3') TCAGCAAACACAGTGCACACCAC;

wild-type forward primer (5' to 3') TTTCCGACTTGAGTTGCCTCAAGA;

wild-type reverse primer (5' to 3') CTGTAGTAAGGATCTCAAGCAGGAGAG.

## Plasmids

For β1–3 gRNAs, the GABRB1#5- GABRB2#2- GABRB3#2 triple gRNA expression unit (U6 promoter + gRNA+ scaffold + PolyT tail) was de novo synthesized by Genscript and cloned into pSpCas9 BB-2A-GFP (β 1-3gRNAs) via AflIII/XbaI sites. To rescue the β1–3 subunit deletion, gRNA-resistant β1*, β2*, β3*, β3*Y302C plasmids were constructed. Briefly, point mutations targeting the β1–3 gRNA sites in β1*, β2*, β3*and the β3*Y302C were generated by overlapping PCR and cloned into the pCAGGS-IRES-mCherry expression plasmid. gRNA-resistant β1*, β2*, β3* or β3*Y302C in pCAGGS-IRES-mCherry, pCAG-Kir2.1-T2A-tdTomato or pCAG-Kir2.1Mut-T2A-tdTomato was co-transfected with β1-3gRNAs. All constructs were verified by DNA sequencing.

## gRNA sequence

The sequence was designed the specific gRNA targets were:

GABRB1 #5 forward (5' to 3') CACCg GCCGCGAGGGCTTCGGGCGT; GABRB1 #5 Reverse (5' to 3') AAAC ACGCCCGAAGCCCTCGCGGC c;

GABRB2 #2 forward (5' to 3') CACCg CAGACAGCGGCGAT TATTAA; GABRB2 #2 reverse (5' to 3') AAAC TTAATAATCG CCGCTGTCTG c;

GABRB3#2 forward (5' to 3') CACCg ACGGTCGACAAGCT GTTGAA; GABRB3#2 reverse (5' to 3') AAAC TTCAACAGCT TGTCGACCGT c.

## Stereotactic injection of AAV

The mice were anesthetized with isoflurane and positioned in a stereotactic instrument equipped with a heating pad (RWD Life Science, Shenzhen, China). For KO of GABA$_A$Rs β1–3 subunits in the hippocampal CA1 region of Rosa26-LSL-Cas9-tdTomato mice, Cre-dependent AAV2/9-β123gRNA-CaMKIIα-EGFP-Cre (β123-CRISPR) or the control virus AAV2/9-CaMKIIα-EGFP-Cre ($1.2 \times 10^{12}$ vg/ml, OBiO Technology, Shanghai, China) was injected into bilateral hippocampus at a rate of 100 nl/min. A total of 0.5 μl of AAV was injected into the dCA1 anteroposterior (AP): −1.7 mm; mediolateral (ML): ±1.5 mm; dorsoventral (DV): −1.5 mm. The incision was sutured, and mice were returned to its home cage for 4-week recovery before subsequent experiments.

## Cell culture and transfection

HEK293T cells were grown in DMEM (GIBCO) supplemented with 10% fetal bovine serum (FBS) (GIBCO), in a humidified atmosphere in a 37 °C incubator with 5% $CO_2$. Transfection was performed in 24-well plates with indicated cDNAs using calcium phosphate transfection.

## Dissociated hippocampal neuronal culture

Hippocampal cultures were prepared from C57BL/6J E18 time-pregnant mice as previously described (Wang et al, 2024). Briefly, the mouse hippocampi were dissected out in ice-cold Hank's balanced salt solution, and digested with papain (Worthington, LK003176) solution at 37 °C for 45 min. After centrifugation for 5 min at 800 rpm, the pellet was resuspended in DNase I-containing Hank's solution, then was mechanically dissociated into single cells by gentle trituration using a pipette. Cells were placed on top of Hank's solution mixed with trypsin inhibitor (10 mg/ml, Sigma T9253) and BSA (10 mg/ml, Sigma A9647), and centrifuged at 800 rpm for 10 min. The pellet was resuspended in Neurobasal plating media containing 2% B27 supplements and GlutaMax (GIBCO). Neurons were plated at a density of 200,000 cells/well on poly-D-lysine (Sigma P6407)-coated 12 mm glass coverslips residing in 24-well plates for electrophysiology recording, and a lower plating density (120,000 cells/well) was adopted when neurons were used for immunocytochemistry. Culture media were changed by half volume with Neurobasal maintenance media containing 2% B27 supplements and GlutaMax (GIBCO) twice a week.

## Neuronal transfection

Hippocampal neurons were transfected at day 3 in vitro (DIV 3) using a modified calcium phosphate transfection. Briefly, 5 μg total cDNA was used to generate 200 μL total precipitates, which was added to each well at a 40 μL volume (5 coverslips/group). After 2 hr incubation in a 37 °C incubator, the transfected cells were incubated with 37 °C pre-warmed, 10% $CO_2$ pre-equilibrated Neurobasal medium, and placed in a 37 °C, 5% $CO_2$ incubator for 20 min to dissolve the calcium-phosphate particles. The coverslips were then transferred back to the original conditioned medium. The cells were cultured to DIV 17–21 or DIV13–16 for experiments.

## Electrophysiology

For recording in acute slices, transverse hippocampal slices (for recording in CA1 pyramidal neurons, 300 μm) were cut from dissected hippocampi on a tissue slicer (Vibratome 3000; Vibratome) in chilled high sucrose cutting solution, containing (in mM): 212.7 sucrose, 3 KCl, 1.25 NaH$_2$PO$_4$, 3 MgCl$_2$, 1 CaCl$_2$, 26 NaHCO$_3$, and 10 glucose, bubbled with 95% O$_2$/5% CO$_2$. Freshly cut slices were placed in an incubating chamber containing carb-oxygenated artificial cerebrospinal fluid (ACSF), containing (in mM) 124 NaCl, 2.5 KCl, 26 NaHCO$_3$, 1 Na$_2$PO4, 10 glucose, 2.5 CaCl$_2$, 1.3 MgCl$_2$ and recovered at 32 °C for ~30 min. All recordings were performed at 28 °C ± 2 °C. Pyramidal neurons in CA1 areas were identified by EGFP fluorescence. Series resistance was monitored and not compensated, and cells in which series resistance varied by 25% during a recording session were discarded. Data was filtered at 1 kHz and digitized at 10 kHz using Igor Pro (WaveMetrics, Portland, OR, USA).

To isolate GABA$_A$R-mediated mIPSCs from pyramidal cells in CA1, 1 μM TTX, 10 μM CNQX, and 50 μM APV were added to the ACSF. GABA$_A$R-mediated sIPSCs were recorded without TTX. To isolate AMPAR-mediated mEPSCs, 1 μM TTX, 100 μM PTX, and 50 μM APV were added to the ACSF. sIPSCs/mIPSCs were recorded at a holding potential (Vh) of –60 mV as inward current with the Cs-based internal solution consisting of the following (in mM): 130 CsCl, 8 NaCl, 0.2 EGTA, 10 HEPES, 5 lidocaine N-ethyl bromide (QX-314), 4 Mg-ATP, and 0.5 Na-GTP, pH 7.4, at 270–290 mOsm. mEPSCs were recorded at a holding potential (Vh) of –70 mV as inward current with internal solution consisting of the following (in mM): 130 Cs-methylsulfonate, 10 HEPES, 10 Na-phosphocreatine, 5 QX-314, 4 Mg-ATP, 0.5 Na-GTP, pH 7.2–7.3. Acquired mEPSCs and mIPSCs were analyzed using the Mini Analysis ProgramTM (Synaptosoft). The threshold for detecting mEPSCs and mIPSCs was set at three times the Root Mean Square (RMS) noise.

Evoked EPSCs were recorded by whole-cell voltage-clamp mode. A concentric bipolar stimulating electrode with a tip diameter of 125 μm (FHC) was placed in the stratum radiatum. The distance between the stimulating and recording electrode was kept at 50–100 μm. Patch pipettes (3–5 MΩ) were filled with the internal solution consisting of the following (in mM): 130 Cs-methylsulfonate, 10 HEPES, 10 Na-phosphocreatine, 5 QX-314, 4 Mg-ATP, 0.5 Na-GTP, pH 7.2–7.3; the osmolarity of the solution was 270–285 mOsm. To obtain NMDAR-EPSCs to AMPAR-EPSCs ratio, AMPAR-EPSC was first recorded in ACSF solution (containing PTX) at –70 mV, and then the same cell was held at +40 mV to record NMDAR-EPSC in the presence of CNQX (10 μM) and PTX (100 μM) with the same stimulus pulse (0.1 ms, 50 μA); the ratio of the maximal amplitude of NMDAR-EPSPCs to AMPAR-EPSCs was defined as the NMDAR/AMPAR ratio. For input–output responses, AMPAR-EPSCs, recorded at –70 mV in the presence of APV (50 μM) and PTX (100 μM), and NMDAR-EPSCs, recorded at +40 mV in the presence of CNQX (10 μM) and PTX (100 μM) were

evoked by a pulse electrical stimulus (0.1 ms width) with different stimulus intensities systematically (5, 10, 15, 20, 25, 30, 40, 60, and 80 µA). Inter-stimulus intervals were >15 s to minimize depression resulting from repetitive stimulation, and at least 10 responses for each intensity were averaged to measure the AMPAR-EPSCs and NMDAR-EPSCs.

To examine the neurotransmitter release probability from presynaptic inputs in CA1 pyramidal neurons, paired-pulse ratio (PPR) of evoked EPSCs at –70 mV was measured at different inter-stimulus intervals (10, 20, 30, 50, 100, 200, 300, 500, 1000 and 2000 ms).

Recordings exhibiting dual-synaptic features were excluded from all analyses.

To induce LTP, a pairing protocol in whole-cell recording mode was applied. In brief, conditioning stimulation consisting of 360 pulses at 2 Hz was paired with continuous postsynaptic depolarization (180 s) to 0 mV. To suppress excessive polysynaptic activity, PTX (100 µM) was added in the recording bath, and the concentration of divalent cations was elevated to 4 mM $Ca^{2+}$ and 4 mM $Mg^{2+}$ to reduce recruitment of polysynaptic responses. A test pulse was delivered at 0.05 Hz to monitor the baseline amplitude for approximately 8 min before, and for about 40 min after, paired stimulation. LTP was calculated by normalizing the EPSC amplitude to the average baseline amplitude over the 8-minute baseline period. Potentiation was defined as the mean normalized EPSC amplitude 20–30 min following paired stimulation. Data were acquired in an interleaved manner for LTP comparisons between β123-CRISPR and control mice.

For current clamping recording, the intracellular solution contained (in mM) KMeSO4 130, KCl 10, HEPES 10, NaCl 4, EGTA 1, Mg-ATP 4, and Na-GTP 0.3. Brain slices or primary neuronal cultures were perfused with standard ACSF saturated with 95% $O_2$/5% $CO_2$.

For dissociated hippocampal neuronal cultures, the intracellular solution for GABA-evoked recording contained (in mM) 130 CsCl, 8 NaCl, 0.2 EGTA, 10 HEPES, 5 lidocaine QX-314, 4 Mg-ATP, and 0.5 Na-GTP, pH 7.4, at 270–290 mOsm. The intracellular solution for AMPA or NMDA-evoked whole-cell currents or NMDA-mEPSC recording contained (in mM) 130 Cs-methylsulfonate, 10 HEPES, 10 Na-phosphocreatine, 5 QX-314, 4 Mg-ATP, 0.5 Na-GTP, pH 7.2–7.3; the osmolarity of the solution was 270–285 mOsm. The external solution for GABA-evoked whole-cell currents contained (in mM) 150 NaCl, 5 KCl, 1MgCl$_2$, 10 HEPES, 2 CaCl$_2$, 10 glucose, 0.05 APV, 0.01 CNQX, and 0.0005 TTX, pH 7.3–7.4. The external solution for AMPA-evoked whole-cell currents contained (in mM) 150 NaCl, 5 KCl, 1MgCl$_2$, 10 HEPES, 2 CaCl$_2$, 10 glucose, 0.05 APV, 0.1 PTX, and 0.0005 TTX, pH 7.3–7.4. The external solution for NMDA-evoked whole-cell currents contained (in mM) 150 NaCl, 5 KCl, 0 mM MgCl$_2$, 10 HEPES, 2 mM CaCl$_2$, 10 glucose, 0.0002 glycine, 0.1 PTX, 0.01 CNQX, and 0.0005 TTX, pH 7.3–7.4. GABA-evoked whole-cell currents were induced by GABA (200 µM) in hippocampal dissociated cultures, a glass pipette filled with external solution containing GABA was placed at the same distances (~150 µm) to a control and EGFP or mCherry-positive neurons at the same time. Pressure pulses (15 psi; 2 s duration) were delivered to eject GABA at 30 s intervals. AMPA or NMDA-evoked whole-cell currents were measured by rapidly applying 200 µM AMPA or NMDA respectively to neurons for 0.5 s every 30 s. The mean current amplitude

of the GABA, AMPA or NMDA responses for a neuron was determined by averaging three successive trials (30 s intervals). NMDA mEPSCs were recorded at +40 mV, 0.0002 glycine, 0.1 PTX, 0.01 CNQX, and 0.0005 TTX (in mM) were added to external solution.

## Morris water maze

Mouse behavioral studies Morris Water Maze (MWM) were used to investigate changes in learning and memory. The device is a circular white pool (120 cm diameter × 50 cm depth) filled with water dyed white with $TiO_2$, and with temperature maintained at 22 °C. A 10-cm-diameter platform was placed 1 cm below the water surface at a fixed position. Mice were trained with four trials per day for 5 consecutive days. Each trial lasted 60 s or until the mouse found the platform. If the mouse did not find the platform during the allocated time period, the experimenter directed the mouse to the platform. After each trial, the mouse was placed on the platform for 10 s. On the 6th day, the platform was removed for a probe trial (60 s) to assess long-term spatial memory retrieval. All parameters were recorded by a video tracking system (Topscan™3.0, Clever Sys., Inc).

## Y-maze working memory

The Y-shaped maze contains three identical arms (34 cm long, 8 cm wide, and 14 cm deep) marked as A, B, and C. Each mouse was placed at the end of one of the three arms randomly to begin the spatial working memory test. The total number and the order of arm entries were recorded during an 8 min test. To count one arm entrance, all four limbs of the mouse must enter the arm. The spontaneous alternation among three arms was defined as non-overlapping entrance sequences (i.e., ABC, ACB, CAB, etc.). The percentage of the spontaneous alternation was calculated according to the following formula: Alternation (%) = [number of actual alternations/maximum number of alternations in theory (total number of arm entries-2)] × 100%.

## PTZ-induced seizure

Male Cas9 mice with hippocampal CA1 AAV-U6-gRNA1-U6-gRNA2-U6-gRNA3-CAMKIIa-GFP-2A-Cre-CW3SL or the control AAV-CAMKIIa-GFP-2A-Cre-CW3SL virus infected 5 weeks later were injected intraperitoneally (i.p.) with PTZ (Sigma-Aldrich, MO, USA, a GABA$_A$ receptor antagonist, dissolved in saline) at a dose of 40 mg per kg of body weight to induce seizures and monitored for 10 min after the injection (Li et al, 2017). Behavioral responses were scored at every 1 min using the following scale: 0, no abnormal behavior; 1, reduced motility and prostrate position; 2, partial clonus; 3, generalized clonus including extremities; 4, tonic-clonic seizure with rigid paw extension; 5, death, as described before (Li et al, 2017). Total seizure scores were calculated by summing up the minute-by-minute scores.

## Drug treatment

D-cycloserine (DCS, Selleck) was dissolved in saline at a stock concentration of 2 mg/ml. A single dose of DCS (20 mg/kg body weight) was intraperitoneally (i.p.) administered to KO mice

90 min before the Y-maze test. Saline was administered at equal volume as a vehicle control. In order to investigate whether DCS affects LTP of synaptic transmission, DCS (25 μM) was added to ACSF 10 min prior to the baseline acquisition and maintained until the end of the experiment.

## Immunocytochemistry

HEK293T cells grown on coverslips were rinsed with PBS twice and fixed in 4% paraformaldehyde (PFA)/4% sucrose/1× PBS solution for 15 min at RT, permeabilized and blocked with 0.1% Triton X-100/10% normal goat serum in 1× PBS for 1 h. Cells were labeled with primary antibodies as follows: anti-β1 (1:500, SYSY-224703), anti-β2 (1:800, Sigma-Aldrich, AB5561), anti-β3 (1:1000, Sigma-Aldrich, SAB2100880) in 3%NGS/1× PBS solutions, incubated overnight at 4 °C. Cells were washed three times with 1× PBS and then incubated with Alexa Fluor 647-conjugated IgG for 1 h. Coverslips were washed four times with 1× PBS and mounted with Fluoromount-G (0100-01, Southern Biotech).

For surface labeling, hippocampal neurons on the coverslips were live-stained with anti-GluN1 antibodies (1:200, SYSY, 114 103) in conditioned media at 37 °C for 1 hr. Coverslips were then washed three times with 1× PBS, and fixed for 15 min with pre-warmed fixation buffer (4% paraformaldehyde and 4% sucrose in 1× PBS, pH 7.4). After washing, the coverslips were incubated with Alexa-conjugated 555 Affini-Pure goat anti-Rabbit (Jackson Immuno Research) secondary antibodies in 1% BSA (in 1× PBS) for 1 h at RT. After washing, coverslips were mounted with DAPI-containing Fluoromount-G.

## Immunohistochemistry

Four weeks after the bilateral hippocampal injection of either AAV2/9-β123gRNA-CaMKIIα-EGFP-Cre or AAV2/9-CaMKIIα-EGFP-Cre, mice were anesthetized with isoflurane, perfused trans-cardially with 4% PFA in 0.1 M phosphate buffer, pH 7.4 (PB). Brains were then dissected out, post-fixed in the same solution for 1 h at RT, and then cryoprotected overnight in 30% sucrose in PB at 4 °C. Coronal brain sections (40 μm) were collected at −20 °C with a cryostat (Leica CM1050). Brain sections were then washed three times (20 min each time) in 1× PBS then mounted on super-frost slides and covered with DAPI-containing Fluoromount-G.

## Image acquisition and analysis

Fluorescence images were obtained with a Zeiss LSM 880 laser scanning confocal microscope with identical settings for laser power, gain, offset, pinhole size, and z-steps throughout the same experiments. The region of interest (ROI) in brain sections was determined using DAPI images.

For image acquisition of Immunohistochemistry staining, serial confocal z stack images were acquired with a resolution of 1024 × 1024, using a ×63 oil objective (numerical aperture 1.4) lens. The maximal intensity projected images were generated by LSM browser software for analysis. Scan speed function was set to 7 and the mean of 4 lines was collected. Pinhole was set to 1 airy unit for all experiments. Laser power, digital gain and offset settings were made identical in each experiment by using the 'reuse' function in LSM software. For puncta analysis, Images from 3 dendrites (35 μm in length) per neuron were collected and quantified with ImageJ puncta analyzer program. Thresholds were set at 3 SDs above the mean staining intensity of six nearby regions in the same visual field. Threshold images display a fixed intensity for all pixels above threshold after removing all of those below it. Labeled puncta were defined as areas containing at least four contiguous pixels after thresholding.

For image acquisition of immunohistochemistry staining, single-plane confocal images were captured with a resolution of 1024 × 1024, using a ×40 oil objective (1.4 numerical aperture).

## Quantification and statistical analysis

All data were presented as mean ± SEM (standard error of mean). Direct comparisons between two groups were made using two-tailed Student's $t$ test. Multiple group comparisons were made using one-way analysis of variance (ANOVA) with Bonferroni test or two-way repeated-measures ANOVA. Statistical significance was defined as $P < 0.05$, 0.01, 0.001 or 0.0001 (indicated as *, **, *** or ****, respectively). n.s., $P$ values $\geq 0.05$ were considered not significant.

## Graphics

SYNOPSIS was created with BioRender.com.

# Data availability

This study includes no data deposited in external repositories.

The source data of this paper are collected in the following database record: biostudies:S-SCDT-10_1038-S44319-025-00538-x.

# Peer review information

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

## Acknowledgements

This work was supported by grants from Guangdong Natural Science Foundation (2023A1515012164) and Advanced Medical Technology Center, The First Affiliated Hospital, Zhongshan School of Medicine, Sun Yat-sen University. Wei Lu was supported by NINDS Intramural Program and is currently supported by SMART Startup Package.

## Author contributions

**Jing-jing Duan**: Conceptualization; Resources; Data curation; Software; Formal analysis; Supervision; Funding acquisition; Validation; Investigation; Visualization; Methodology; Writing—original draft; Project administration; Writing—review and editing. **Bin Jiang**: Resources; Methodology; Writing—review and editing. **Wei Yin**: Resources; Writing—review and editing. **Yuan Lin**: Funding acquisition; Writing—review and editing. **Guang-mei Yan**: Resources; Investigation; Writing—review and editing. **Wei Lu**: Conceptualization; Investigation; Writing—review and editing.

Source data underlying figure panels in this paper may have individual authorship assigned. Where available, figure panel/source data authorship is listed in the following database record: biostudies:S-SCDT-10_1038-S44319-025-00538-x.

## Disclosure and competing interests statement

The authors declare no competing interests.

# Expanded View Figures

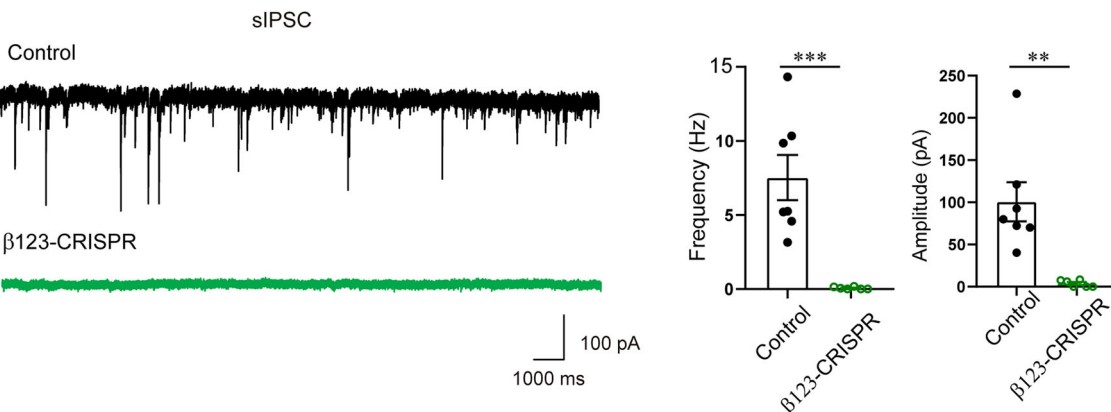

**Figure EV1. Genetic deletion of GABA_AR β1, β2, and β3 subunits via β123-CRISPR eliminated spontaneous IPSCs in hippocampal CA1 neurons expressing β123-CRISPR compared to control, related to Fig. 1.**

Bar graphs indicate mean ± SEM, $n = 7$ for control, $n = 6$ for β123-CRISPR; ***$P = 0.0009$, **$P = 0.0028$, unpaired $t$ test.

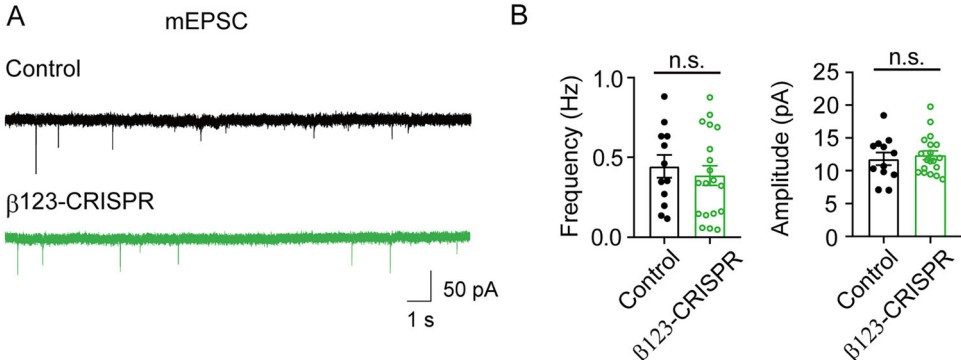

**Figure EV2.    Neither the amplitude nor the frequency of AMPAR-mediated mEPSCs showed a significant change in neurons expressing β123-CRISPR, related to Fig. 2.**

(A) Representative traces of AMPAR-mediated mEPSCs recorded from control and β123-CRISPR expressing neurons. (B) Quantification of mEPSC amplitude and frequency. Bar graphs indicate mean ± SEM. Frequency: control vs β123-CRISPR, 0.44 ± 0.07 Hz vs 0.39 ± 0.06 Hz, $P = 0.55$; Amplitude: control vs β123-CRISPR, 11.79 ± 0.96 pA vs 12.34 ± 0.67 pA, $P = 0.63$; $n = 12, 19$, $N = 3$ for each group, n.s., not significant, unpaired $t$ test.

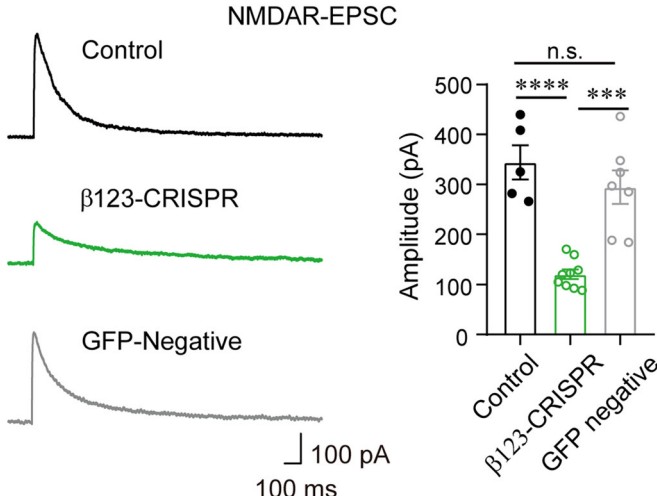

**Figure EV3.   NMDAR transmission remains unaltered in GFP-negative neurons in β123-CRISPR mice, related to Fig. 2.**

Bar graphs indicate mean ± SEM, $n = 5$ for control, $n = 9$ for β123-CRISPR, $n = 7$ for GFP-negative neurons in β123-CRISPR mice; ***$P = 0.0002$, ****$P < 0.0001$, n.s., not significant, $P = 0.5105$, one-way AVOVA with Bonferroni's test.

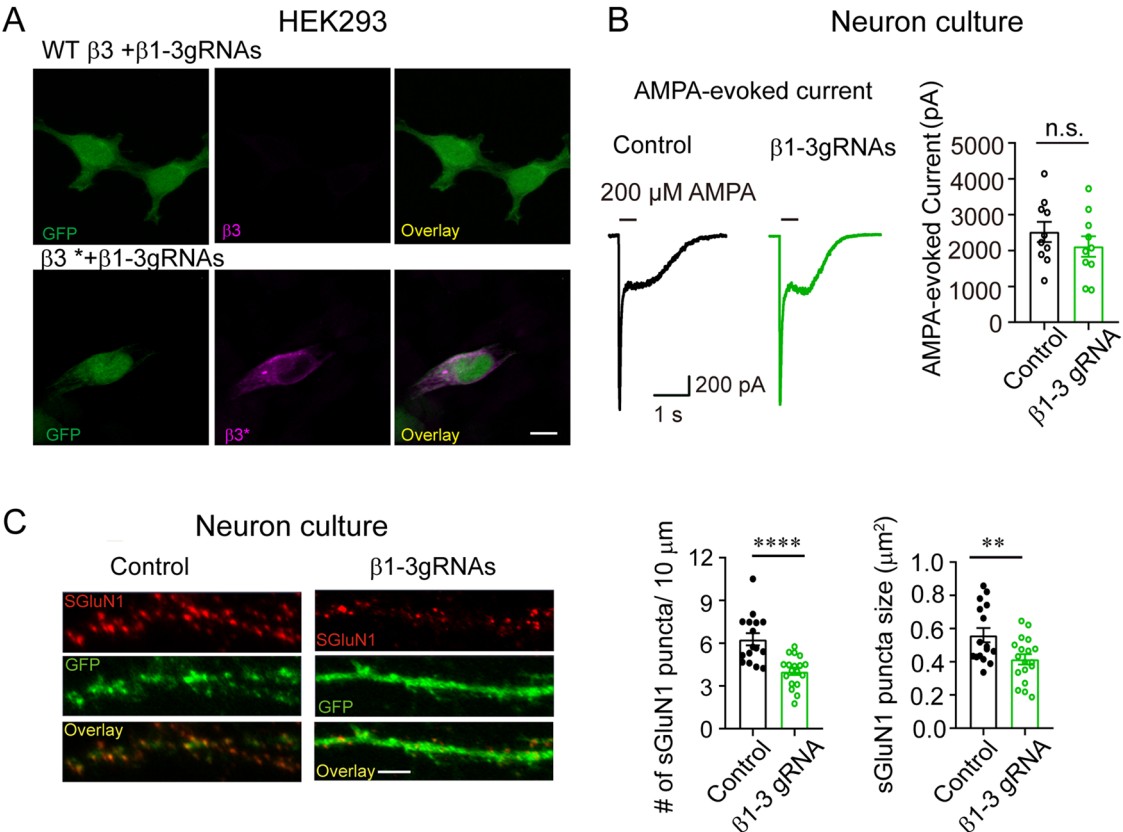

**Figure EV4.** **β1–3 gRNA maintains gRNA-Resistant β3 (β3*) expression, preserves AMPA-evoked currents, but reduces surface GluN1, related to Fig. 3.**

(A) Representative images showed that β1–3 gRNA failed to diminish the expression of the gRNA-resistant-β 3* (β3*) in HEK293T cells. Scale bar, 2 μm. (B) Loss of GABA_ARs in neurons expressing β1–3 gRNAs didn't significantly affect AMPA-evoked whole-cell currents in hippocampal neuronal cultures (Bar graphs indicate mean ± SEM, $n = 10$, $N = 3$, $P = 0.3244$, unpaired $t$ test). Bars indicate AMPA applications for AMPA-evoked whole-cell currents. (C) Loss of GABA_ARs in neurons expressing β1–3 gRNAs significantly reduced surface GluN1 puncta (red) in hippocampal neuronal cultures. Representative images of surface GluN1 in neurons expressing control-GFP or β1–3 gRNAs (left). (Right) Bar graphs showed the quantitation of surface GluN1 puncta density and size in neuronal dendrites. Bar graphs indicate mean ± SEM, $n = 16$, 18 respectively, $N = 3$, ****$P < 0.0001$, **$P = 0.0089$, unpaired $t$ test). Scale bar, 5 μm.

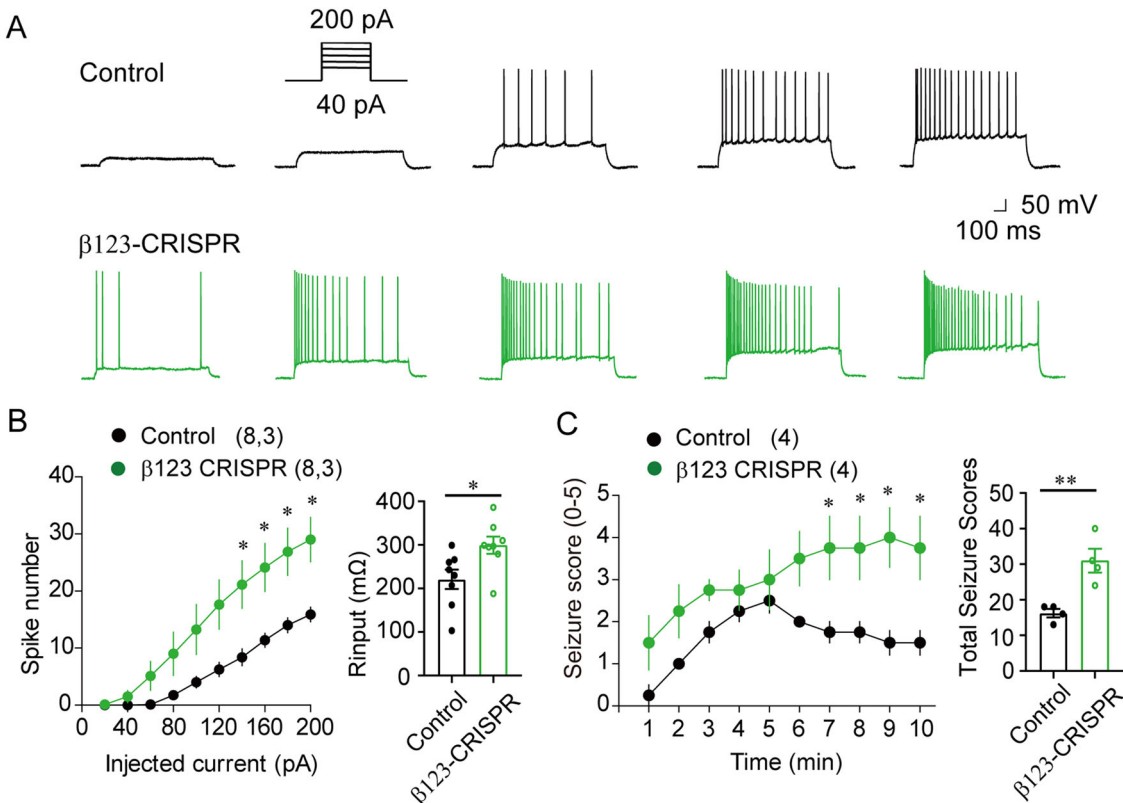

**Figure EV5. Genetic deletion of GABA_ARs in β123-CRISPR mice induced enhanced neuronal excitability and hippocampal hyperexcitability, related to Fig. 4.**

(A) Sample action potential responses to step current injections (40, 80, 120, 160 and 200 pA; 1,000 ms) in control neurons (black) or β123-CRISPR neurons (green). (B) Left: summary graph showing that deletion of GABA_ARs increased the excitability of CA1 pyramidal neurons (Line graphs indicate mean ± SEM, *$P = 0.0324$, $n = 8$, $N = 3$, two-way repeated-measures ANOVA followed by Sidak's multiple comparisons test at each intensity). Right: Bar graph shows that input resistance is significantly enhanced in neurons lacking GABA_ARs. *$P = 0.02$, $n = 8$. unpaired $t$ test, Error bars represent SEM. (C) Time course of average seizure scores induced by PTZ injection (40 mg/kg) (Line graphs indicate mean ± SEM, *$P = 0.01$, $n = 4$ male mice/group, unpaired $t$-test). Behavioral responses were scored every 1 min for 10 min after the PTZ injection. Bar graph showing total seizure scores (Bar graphs indicate mean ± SEM, **$P = 0.0059$, $n = 4$, unpaired $t$ test).

