## [Peer Review File · EMBO Reports]

Cell-autonomous GABAARs are essential for NMDAR-mediated synaptic transmission, LTP, and spatial memory

Jing duan, Bin Jiang, Wei Yin, Yuan Lin, Guangmei Yan, and Wei Lu

Corresponding author(s): Jing duan (duanjj2@mail.sysu.edu.cn)

Review Timeline:

Submission Date:	29th Jan 25
Editorial Decision:	24th Feb 25
Revision Received:	26th May 25
Editorial Decision:	7th Jul 25
Revision Received:	8th Jul 25
Accepted:	17th Jul 25

Editor: *Esther Schnapp*

Transaction Report:

Dear Dr. Duan,

Thank you for the submission of your manuscript to EMBO reports. We have now received the full set of referee reports that is pasted below.

As you will see, the referees acknowledge that the findings are interesting. However, they also have several suggestions for how the study should be strengthened, and I think all suggestions are good and should be addressed. Please let me know in case you disagree, and we can discuss the exact revision requirements further, also in a video chat, if you like.

I would thus like to invite you to revise your manuscript with the understanding that the referee concerns must be fully addressed and their suggestions taken on board. Please address all referee concerns in a complete point-by-point response. Acceptance of the manuscript will depend on a positive outcome of a second round of review. It is EMBO reports policy to allow a single round of major revision only and acceptance or rejection of the manuscript will therefore depend on the completeness of your responses included in the next, final version of the manuscript.

We realize that it is difficult to revise to a specific deadline. In the interest of protecting the conceptual advance provided by the work, we recommend a revision within 3 months (27th May 2025). Please discuss the revision progress ahead of this time with the editor if you require more time to complete the revisions.

- 1) A data availability section providing access to data deposited in public databases is missing. If you have not deposited any data, please add a sentence to the data availability section that explains that.
- 2) Your manuscript contains statistics and error bars based on $n=2$. Please use scatter blots in these cases. No statistics should be calculated if $n=2$.

5) a complete author checklist, which you can download from our author guidelines . Please insert information in the checklist that is also reflected in the manuscript. The completed author checklist will also be part of the RPF.

6) Please note that all corresponding authors are required to supply an ORCID ID for their name upon submission of a revised

manuscript (). Please find instructions on how to link your ORCID ID to your account in our manuscript tracking system in our Author guidelines

- the name of the statistical test used to generate error bars and P values,
- the number (n) of independent experiments (please specify technical or biological replicates) underlying each data point,
- the nature of the bars and error bars (s.d., s.e.m.),
- If the data are obtained from n {less than or equal to} 2, use scatter blots showing the individual data points.

12) All Materials and Methods need to be described in the main text using our 'Structured Methods' format, which is required for all research articles. According to this format, the Methods section includes a Reagents and Tools Table (listing key reagents, experimental models, software and relevant equipment and including their sources and relevant identifiers) followed by a Methods and Protocols section describing the methods using a step-by-step protocol format. The aim is to facilitate adoption of the methodologies across labs. More information on how to adhere to this format as well as a downloadable template (.docx) for the Reagents and Tools Table can be found in our author guidelines:

An example of a Method paper with Structured Methods can be found here: <https://www.embopress.org/doi/full/10.1038/s44320-024-00037-6#sec-4>

As part of the EMBO publication's Transparent Editorial Process, EMBO reports publishes online a Review Process File (RPF) to accompany accepted manuscripts. This File will be published in conjunction with your paper and will include the referee

reports, your point-by-point response and all pertinent correspondence relating to the manuscript.

I look forward to seeing a revised form of your manuscript when it is ready.

Referee #1:

This manuscript describes studies on the cell-autonomous role of GABA-A receptors in regulating glutamatergic neurotransmission. The authors use the CRISRP/Cas-9 system to knock down all three GABA-A receptor beta subunits (beta1-

3) in excitatory neurons in the hippocampal CA1 region, performing studies in hippocampal slices and cell cultures. To my knowledge, this is the first example where a functional knockout of apparently all GABA-A receptors in a particular neuron subtype is being described. This knockdown led to a reduction of NMDA receptor-mediated, but not of AMPA receptor-mediated synaptic transmission, including a loss of long-term potentiation (LTP) and impaired spatial memory. The authors also show that the effect of the GABA-A receptor system can be rescued by the beta2 and beta3 subunits, but only partially by the beta1 subunit, and also not be a the beta3(Y302C) mutant subunit. D-cycloserine rescued essential deficits, indicating that NMDA receptor hypofunction induced by loss of GABA-A receptor function can be rescued by a compound acting at the co-agonist binding site of the NMDA receptor. The research described in this manuscript is highly innovative, and provides an important expansion of our knowledge of how GABA-A receptors and NMDA receptors interact to modulate the balance between excitation and inhibition. Moreover, the manuscript is written exceptionally well, and as a result, I have only a very small number of minor comments. The paper will be a very welcome addition to scientific literature.

Minor comments

Page 4, line 6: It should probably be "odds", not "odd".

Page 6, lines 7-8: the phrase "to this end" does not make a lot of sense here. Consider omitting it.

Page 9, lines 28-29: "while the mice were traveling the similar distance during the test (Figure 5B-D)". It should be "traveling a similar distance". Strictly speaking, Fig. 5E shows swimming speed and not distance traveled. It would perhaps make more sense to show the distance traveled in the figure.

Page 11, line 29: "can occur independent" should be "can occur independently".

Page 34, line 19: "downward" is not really the ideal word here.

Page 36, line 17: Spell out "DCS" here. Figure legends should be self-sufficient.

Page 39, Figure 3C,D,F. The authors formally compare all values just to control. I would have liked to see a statistical analysis of the differences between "beta1-3gRNA" and all the potential rescue group. For example, it looks like that the beta1 subunit leads to a very substantial increase in GABA-evoked current (in Fig. 3C), for which statistical analysis is apparently not performed.

Referee #2:

In this study, Duan et al. investigate the role of GABAARs in NMDAR-mediated transmission and LTP expression in hippocampal pyramidal neurons, as well as in memory tasks which require these synaptic mechanisms. They perform triple KO of all three beta subunits in a subset of CA1 pyramidal neurons, which almost eliminates all spontaneous inhibitory inputs. This manipulation leads to a stark reduction in NMDAR-, but not AMPAR-mediated EPSC amplitudes and thus dampens NMDA/AMPA ratio, which appears to be a postsynaptic effect as the paired-pulse ratio remains unchanged. To establish a causal relationship between the GABAR KO and NMDA transmission reduction, they perform rescue experiments in cultured hippocampal neurons by co-expressing the KO vector with specific beta subunits. They replicate the slice results here by showing that GABAR KO also results in diminished NMDAR transmissions, which strikingly can be rescued by the co-expression of these beta subunits. They further show that this diminished NMDAR activation can be restored by introducing Kir2.1 channel, which renders the neuron less excitable, revealing that this downregulation of NMDAR activity could be a homeostatic mechanism triggered by elevated excitation. Finally, they show that LTP is abolished in the KO neurons, and animals that have undergone this type of manipulation perform poorly in memory tasks strongly correlated with NMDAR function and LTP expression. Strikingly both the LTP expression and task performance can be rescued by pharmacological interventions that

promote NMDAR function.

This valuable study establishes a causal relationship between GABAAR function and NMDAR-mediated transmission in CA1 pyramidal neurons, revealing an important molecular mechanism that contributes to the maintenance of overall E-I balance. Most experiments are appropriately designed, the data are clearly illustrated and the results are convincing. Importantly, the implication of this molecular interaction in memory tasks highlights the functional significance of the study. I believe these findings will be of substantial interest to investigators in the synaptic plasticity and/or E/I balance fields. Despite these strengths, certain claims are not sufficiently supported by the data presented in the current study. Below I list my questions and concerns for the authors.

Major concerns:

1. A major claim the authors try to make is that this regulation of NMDAR transmission is cell-autonomous, and they attempt to support this by doing sparse KO in CA1. The sample image in Fig. 1D does not suggest the expression is sparse (also the right image does not match the boxed area in the left one, max projection vs. a single step?). With this expression rate, it is difficult to rule out a network effect (aka by the overall increase in network activity due to loss of inhibition in many excitatory cells). Have the authors quantified the expression rate of the KO vector in CA1? Further, the authors state that lack of mIPSCs in KO neurons allows for cell-autonomous investigation (Fig. 1E), yet these neurons are still under the influence of other types of synaptic inputs (e.g. feedback excitation through another KO neuron).

For these reasons, in order to argue that this effect is cell-autonomous, the authors should provide evidence that 1) no significant activity change on the network level (such as cFos activity), 2) GFP-negative neurons in the injected area do not exhibit the same phenotype (aka NMDAR transmission should not be affected).

2. Can authors confirm that the reduction in mIPSC frequency in KO neurons is not caused by reduced GABA release probabilities? Given the expression rate illustrated in Fig 1D, I suspect that the activity of nearby interneurons is also affected. I suggest that the authors provide some validation for this KO vector (e.g. staining against GABARs in cultured neurons with or without KO).

3. For experiments performed in Fig 3 and S3, it is not clear in the text which are performed in HEK cells and which are in neurons. The authors mention in the methods that drugs are applied at the same distances from the cell (the soma?), then it is perplexing that AMPA currents in Fig S3B decay more slowly than NMDA currents in Fig 3D.

4. For the section "cell-intrinsic excitability governs NMDAR-mediated synaptic transmission in neurons lacking GABARs" (from p7), the authors propose a model where GABAR loss elevates neuronal excitability, which in turn triggers a homeostatic reduction of NMDAR activity. This model is fine, but it is important to note that "neuronal excitability" and "intrinsic excitability" are two different things. Intrinsic excitability is property measured in the absence of synaptic inputs, whereas neuronal excitability is not. This wording in the section title may be an unintentional mistake, as the authors mostly use "neuronal excitability" in their description of the results here. Otherwise, since the authors have not described how these current clamp recordings were done in the methods, the usage of "intrinsic excitability" is inaccurate here. It is true that Kir2.1 expression can dampen intrinsic excitability, but at the same time it does the same thing to the neuronal excitability. In Fig. 3F, Kir2.1 expression in control neurons also leads to an increase in the NMDAR-mEPSC amplitude, so this NMDAR homeostatic compensation triggered by downregulation of activity is perhaps universal, not just for neurons without GABARs.

I suggest that the authors fully describe these experiments in the methods, and rephrase the corresponding texts to reflect what the data shows (both in results and in discussion).

Minor concerns:

1. P3, lines 2-3, this statement needs citation.

2. P3, line 24, "Excitation-Inhibition (E/I) balance".

3. P4, lines 5-7, I wouldn't say that the prominent E/I balance theory is the balance of AMPA EPSCs and GABA IPSCs. The consensus in the field, including the point discussed in Turrigiano 2012, is that E/I balance results from the coordination of multiple excitatory and inhibitory elements. AMPARs may have been studied more extensively than other channels (e.g. NMDARs), but that doesn't mean it's the only excitatory player here. Similar issues in discussion (P12, lines 18-19): it's more accurate to say, "a new molecular mechanism that contributes to E/I balance".

4. P4, line 13, "whether NMDAR-mediated activity (or transmission) is regulated by ..."

5. The two traces in Fig S1 seem to have different scales on Y axis (baseline), are these recordings obtained under similar settings? The legend is a bit ambiguous; are traces shown here the average waveforms of recordings from all 6 cells, or are they just representative traces from a given cell?

6. P4, line 28, "the mice were perfused and their brain sections were ..."

7. In Fig 2B, could the authors explain why NMDAR-EPSCs do not differ between WT and KO at low intensities? Would NMDA/AMPA ratios also be similar within this range? Does this indicate that this effect on NMDAR transmission is input-dependent?

8. Could the authors explain why data in Fig 2B are subjected to two-way ANOVA tests? It appears that only the amplitudes at the highest intensity are compared, what's the second factor here? If the input intensity is treated as the second factor, then a

two-way ANOVA for repeated measurements should be used. If so then this should be clarified in the text or legend. (Same for PPRs in Fig 2E)

9. Fig 3F is not called in the text.

10. Fig S3A is not called in the text; the legend indicates these are done in HEK cells, as mentioned in above, could the authors provide rationales for using HEK and/or neuronal culture?

11. Data in Fig S4B are similar to those in Fig 2B (both are illustrating repeated input-output measurements), could the authors explain why they choose to use an unpaired t-test here (rather than a two-way ANOVA)?

12. In Fig S5A, if the authors intend to show the same current injection can evoke more spikes in certain conditions, then they should pick traces evoked by the same current step. Also it is not clear what parameter is being compared in Fig S5B.

13. In methods (P19, line 15), it's curious that the authors perform all recordings at 28C rather than at physiological temperatures, is there a specific reason for this choice? It's been reported that both the amplitude and frequency of NMDAR mEPSCs are temperature-sensitive (PMID: 18068304), showing substantial differences between 25C and 35C. Could some of the low frequencies in Fig 3F be caused by the recording temperature?

14. Also in methods (P19, line 18), the authors state that data were filtered at 2 kHz and digitized at 10 kHz, does this include the evoked EPSCs (e.g. Fig 2)? I'm a bit concerned that the fast EPSC rising phase cannot be fully captured with a 2 kHz filter (if digitized at 10 k then the filter band should not be lower than 5k).

Referee #3:

1. The term "single cell" used by the authors is confusing. It appears that "single cell" actually refers to a "single cell type," specifically CA1 pyramidal neurons. Clarifying this terminology would improve readability and accuracy.

2. In Figure 1E, the rightmost graph shows the amplitude of mIPSCs despite the frequency of mIPSCs being nearly abolished. Could this discrepancy be due to incomplete knockout efficiency?

3. In Figure 2C, the authors state that "AMPA-mediated EPSCs were slightly but not significantly decreased compared to control neurons." However, the data show a notable reduction in AMPAR-EPSCs in the B123-CRISPR condition. Additionally, the authors employed a two-way ANOVA for statistical analysis, but a repeated measures two-way ANOVA would be more appropriate for these experiments. Reanalyzing the data using the correct statistical method is recommended.

4. In Figure 2D, the representative traces for B123-CRISPR neurons display dual synaptic inputs to the cell. It is unclear whether dual-synaptic traces are suitable for the analysis presented. The authors should justify the inclusion of these traces or reconsider their use.

5. In Figure 3D, only the Control group exhibited a washout effect. Did the sgRNA-resistant GABAARs used for recovery display different kinetics compared to native GABAARs? If so, this difference should be addressed.

6. While the authors convincingly demonstrate that reduced NMDARs impair SC-CA1 LTP and its associated memory behavior, the connection between the lack of GABAARs and decreased NMDAR function remains unclear. Although Figure 3 shows a GABAAR recovery experiment, this only establishes correlations, not mechanisms. What is the putative mechanism underlying this phenomenon? Further investigation or discussion is warranted.

7. In Figure 5B, the representative trace of B123-CRISPR mice in the Morris water maze suggests they follow the borders of the maze. Did all B123-CRISPR mice exhibit a similar behavior, or is this trace an isolated example? Providing more comprehensive data would strengthen this conclusion.

8. The authors have inconsistently presented their data-some graphs include individual data points, while others display only mean {plus minus} SEM values. What is the rationale behind this discrepancy? Uniform representation of data would enhance clarity and transparency.

9. The authors used DCS to assess whether partial activation of NMDARs could rescue impaired working memory. Did the DCS treatment also include mice subjected to the Morris water maze test? Clarifying this experimental design is important for interpretation.

Dear Editors and Reviewers,

Thank you for your letter and the referees' comments concerning our manuscript entitled “Cell-autonomous GABA_ARs are essential for NMDAR synaptic transmission, LTP, and spatial memory” (Manuscript Number: EMBOR-2025-61250V1). We greatly appreciate the valuable and insightful feedback provided by the reviewers. Their comments are instrumental in guiding us to improve the manuscript.

We would like to proceed with publication as a full article. Accordingly, we have revised the manuscript to align with the formatting requirements for a full article, including maintaining separate Results and Discussion sections and expanding the number of main figures to six, along with five Expanded View Figures.

We have carefully considered each comment and made significant efforts to address them during the revision. Below, we provide a point-by-point response to the reviewers' comments in blue, detailing the changes and improvements we have made.

Referee #1:

This manuscript describes studies on the cell-autonomous role of GABA-A receptors in regulating glutamatergic neurotransmission. The authors use the CRISPR/Cas-9 system to knock down all three GABA-A receptor beta subunits (beta1-3) in excitatory neurons in the hippocampal CA1 region, performing studies in hippocampal slices and cell cultures. To my knowledge, this is the first example where a functional knockout of apparently all GABA-A receptors in a particular neuron subtype is being described. This knockdown led to a reduction of NMDA receptor-mediated, but not of AMPA receptor-mediated synaptic transmission, including a loss of long-term potentiation (LTP) and impaired spatial memory. The authors also show that the effect of the GABA-A receptor system can be rescued by the beta2 and beta3 subunits, but only partially by the beta1 subunit, and also not by the beta3(Y302C) mutant subunit. D-cycloserine rescued essential deficits, indicating that NMDA receptor hypofunction induced by loss of GABA-A receptor function can be rescued by a compound acting at the co-agonist binding site of the NMDA receptor. The research described in this manuscript is highly innovative, and provides an important expansion of our knowledge of how GABA-A receptors and NMDA receptors interact to modulate the balance between excitation and inhibition.

Moreover, the manuscript is written exceptionally well, and as a result, I have only a very small number of minor comments. The paper will be a very welcome addition to scientific literature.

Minor comments

Page 4, line 6: It should probably be "odds", not "odd".

The term "odd" has been corrected to "odds" as suggested in page 4 line 6.

Page 6, lines 7-8: the phrase "to this end" does not make a lot of sense here. Consider omitting it.

The phrase "to this end" has been removed for clarity, as recommended in page 6 line 20.

Page 9, lines 28-29: "while the mice were traveling the similar distance during the test (Figure 5B-D)". It should be "traveling a similar distance". Strictly speaking, Fig. 5E shows swimming speed and not distance traveled. It would perhaps make more sense to show the distance traveled in the figure.

Thank you for the suggestion. We have replaced the swimming speed data with total distance traveled ($p = 0.138$) in the revised Figure 5F. Additionally, we have updated Figure 5D and E to show both the number of crossings and the time spent in the target quadrant during the probe test (** $p = 0.0013$).

Page 11, line 29: "can occur independent" should be "can occur independently".

The phrase "can occur independent" has been revised to "can occur independently" as suggested in page 12 line 29.

Page 34, line 19: "downward" is not really the ideal word here.

We have replaced "downward" with "reduced" to improve the clarity and accuracy of the description in page 35 line 18.

Page 36, line 17: Spell out "DCS" here. Figure legends should be self-sufficient.

As recommended, we have spelled out 'DCS' as 'D-cycloserine (DCS)' to ensure that the figure legend is self-contained and clear to the reader in page 38 line 2.

Page 39, Figure 3C, D, F. The authors formally compare all values just to control. I would have liked to see a statistical analysis of the differences between "beta1-3gRNA" and all the potential rescue group. For example, it looks like that the beta1 subunit leads to a very substantial increase in GABA-evoked current (in Fig. 3C), for which statistical analysis is apparently not performed.

We appreciate the reviewer's insightful comment. We have now performed additional statistical analyses comparing " β 1-3gRNA" to each of the potential rescue groups in page 41. Similar statistical analyses have also been performed for Figure 3D and 4D (page 41 and 42), as recommended. These revisions confirm the robustness of our original findings and do not alter the overall conclusions of the study.

Referee #2:

In this study, Duan et al. investigate the role of GABAARs in NMDAR-mediated transmission and LTP expression in hippocampal pyramidal neurons, as well as in memory tasks which require these synaptic mechanisms. They perform triple KO of all three beta subunits in a subset of CA1 pyramidal neurons, which almost eliminates all spontaneous inhibitory inputs. This manipulation leads to a stark reduction in NMDAR-, but not AMPAR-mediated EPSC amplitudes and thus dampens NMDA/AMPA ratio, which appears to be a postsynaptic effect as the paired-pulse ratio remains unchanged. To establish a causal relationship between the GABAR KO and NMDA transmission reduction, they perform rescue experiments in cultured hippocampal neurons by co-expressing the KO vector with specific beta subunits. They replicate the slice results here by showing that GABAR KO also results in diminished NMDAR transmissions, which strikingly can be rescued by the co-expression of these beta subunits. They further show that this diminished NMDAR activation can be restored by introducing Kir2.1 channel, which renders the neuron less excitable, revealing that this downregulation of NMDAR activity could be a homeostatic mechanism triggered by elevated excitation. Finally, they show that LTP is abolished in the KO neurons, and animals that have undergone this type of manipulation perform poorly in memory tasks strongly correlated with NMDAR function and LTP expression. Strikingly both the LTP expression and task performance can be rescued by pharmacological interventions that promote NMDAR function.

This valuable study establishes a causal relationship between GABAAR function and NMDAR-mediated transmission in CA1 pyramidal neurons, revealing an important molecular mechanism that contributes to the maintenance of overall E-I balance. Most experiments are appropriately designed, the data are clearly illustrated and the results are convincing. Importantly, the implication of this molecular interaction in memory tasks highlights the functional significance of the study. I believe these findings will be of substantial interest to investigators in the synaptic plasticity and/or E/I balance fields. Despite these strengths, certain claims are not

sufficiently supported by the data presented in the current study. Below I list my questions and concerns for the authors.

Major concerns:

1. A major claim the authors try to make is that this regulation of NMDAR transmission is cell-autonomous, and they attempt to support this by doing sparse KO in CA1. The sample image in Fig. 1D does not suggest the expression is sparse (also the right image does not match the boxed area in the left one, max projection vs. a single step?). With this expression rate, it is difficult to rule out a network effect (aka by the overall increase in network activity due to loss of inhibition in many excitatory cells). Have the authors quantified the expression rate of the KO vector in CA1? Further, the authors state that lack of mIPSCs in KO neurons allows for cell-autonomous investigation (Fig. 1E), yet these neurons are still under the influence of other types of synaptic inputs (e.g. feedback excitation through another KO neuron).

For these reasons, in order to argue that this effect is cell-autonomous, the authors should provide evidence that 1) no significant activity change on the network level (such as cFos activity), 2) GFP-negative neurons in the injected area do not exhibit the same phenotype (aka NMDAR transmission should not be affected).

We appreciate the reviewer's thoughtful feedback and concerns regarding the potential contribution of network-level effects to our interpretation of cell-autonomous NMDAR regulation. We agree that distinguishing cell-autonomous changes from broader network alterations is critical for accurate interpretation. To address the reviewer's points:

1. We agree that the original image in Figure 1D does not clearly support sparse expression. Accordingly, we have removed the term "sparse" and revised the description in the manuscript (page 5, line 1) to more accurately reflect the observed expression pattern.
2. As suggested, we recorded NMDAR-mediated EPSCs from three groups of hippocampal CA1 neurons: GFP-positive neurons in control mice, and both GFP-positive and GFP-negative neurons in β 123-CRISPR mice. While GFP-positive neurons expressing β 123-CRISPR showed a significant reduction in NMDAR-EPSCs, GFP-negative neurons from the same regions did not differ significantly from control GFP-positive neurons (Figure EV3). These results support the conclusion that the observed reduction in NMDAR transmission is specific to β 123-CRISPR expressing neurons and not due to a generalized network effect. The new data are presented in the revised Figure EV3, and page 5, line 28 of the revised manuscript.

3. Assessment of network activity using c-Fos: To examine potential changes in global neuronal activity, we performed c-Fos immunostaining. We found no significant difference in c-Fos expression between β 123-CRISPR and control mice. Moreover, c-Fos-positive cells did not consistently colocalize with β 123-CRISPR-expressing neurons. We acknowledge that c-Fos is a transient marker of neuronal activity, typically detectable 20–90 minutes after stimulation, with a half-life of approximately two hours (<https://doi.org/10.3390/neurosci3040050>). While c-Fos is effective for detecting acute activity changes, it may not reliably capture chronic or subtle alterations in network activity (more than 4 weeks in our case). Therefore, the lack of c-Fos signal changes between β 123-CRISPR and control mice may reflect either the method's limited temporal resolution or a true absence of heightened network activation. Given these limitations, we have opted not to include the c-Fos data in the manuscript. However, we are happy to include these results upon the reviewer's request.

In summary, although subtle network-level effects cannot be entirely ruled out, recordings from neighboring GFP-negative neurons, along with whole-cell recordings of NMDA-evoked currents, NMDA-mEPSCs, and surface GluN1 puncta in primary hippocampal cultures, support the conclusion that the observed reduction of NMDAR transmission in neurons devoid of GABA_ARs is unlikely due to a generalized network effect. The manuscript has been updated accordingly.

2. Can authors confirm that the reduction in mIPSC frequency in KO neurons is not caused by reduced GABA release probabilities? Given the expression rate illustrated in Fig 1D, I suspect that the activity of nearby interneurons is also affected. I suggest that the authors provide some validation for this KO vector (e.g. staining against GABARs in cultured neurons with or without KO).

We thank the reviewer for raising this point. Due to the near-complete elimination of IPSCs in GABA_AR knockout neurons, paired-pulse ratios (PPRs) could not be reliably measured in our recordings. However, the efficacy of the GABA_AR KO vector was previously validated through immunostaining and electrophysiology in primary cultured neurons, which confirmed the loss of β 1, β 2, and β 3 subunit expression and the absence of mIPSCs (Duan et al., 2019, <https://doi.org/10.3389/fncel.2019.00217>). In the present study, we further show that deletion of these β subunits abolished GABA-evoked whole-cell currents. *In vivo*, the elimination of GABA_ARs in hippocampal CA1 pyramidal neurons led to the silencing of GABAergic transmission, underscoring the specificity and effectiveness of the KO strategy.

This clarification has been added to the Discussion section of the manuscript (Page 11, Line 6).

3. For experiments performed in Fig 3 and S3, it is not clear in the text which are performed in HEK cells and which are in neurons. The authors mention in the methods that drugs are applied at the same distances from the cell (the soma?), then it is perplexing that AMPA currents in Fig S3B decay more slowly than NMDA currents in Fig 3D.

We appreciate the reviewer's observation. The distinction between experiments conducted in HEK cells and those performed in neurons has now been clarified in the legends of Figure 3 and 4 (pages 41–42), as well as Figure EV4.

Regarding the concern about the decay kinetics of AMPA currents, we agree that this discrepancy warranted further investigation. To ensure consistent drug delivery conditions, we repeated the experiment using direct application of AMPA (abcam, Cat#144483). The original trace has been replaced with the updated recording. These revised results have been incorporated into Figure EV4B, and the corresponding figure legend has been updated accordingly (page 45, line 20). Importantly, the conclusion remains unchanged: AMPAR-mediated whole-cell currents were not significantly altered in neurons expressing $\beta 1-3$ gRNAs. These revisions reinforce the robustness of our original findings and do not affect the overall conclusions of the study.

4. For the section "cell-intrinsic excitability governs NMDAR-mediated synaptic transmission in neurons lacking GABARs" (from p7), the authors propose a model where GABAR loss elevates neuronal excitability, which in turn triggers a homeostatic reduction of NMDAR activity. This model is fine, but it is important to note that "neuronal excitability" and "intrinsic excitability" are two different things. Intrinsic excitability is property measured in the absence of synaptic inputs, whereas neuronal excitability is not. This wording in the section title may be an unintentional mistake, as the authors mostly use "neuronal excitability" in their description of the results here. Otherwise, since the authors have not described how these current clamp recordings were done in the methods, the usage of "intrinsic excitability" is inaccurate here. It is true that Kir2.1 expression can dampen intrinsic excitability, but at the same time it does the same thing to the neuronal excitability. In Fig. 3F, Kir2.1 expression in control neurons also leads to an increase in the NMDAR-mEPSC amplitude, so this NMDAR homeostatic compensation triggered by downregulation of activity is perhaps universal, not just for neurons without GABARs.

I suggest that the authors fully describe these experiments in the methods, and rephrase the corresponding texts to reflect what the data shows (both in results and in discussion).

We thank the reviewer for pointing out the distinction between “neuronal excitability” and “intrinsic excitability.” We agree that the term “intrinsic excitability” was used inaccurately in this context, as our recordings were not performed in synaptic isolation. Accordingly, we have replaced “intrinsic excitability” with “neuronal excitability” throughout the relevant sections of the manuscript to more accurately reflect the experimental conditions and interpretations. The figure legend for Figure 4 (previously Fig. 3F) has also been updated accordingly.

In addition, we have now updated the Methods section to describe the current-clamp recordings. Specifically, the intracellular solution used contained (in mM): 130 KMeSO₄, 10 KCl, 10 HEPES, 4 NaCl, 1 EGTA, 4 Mg-ATP, and 0.3 Na-GTP. Brain slices or primary neuronal cultures were continuously perfused with standard ACSF saturated with 95% O₂/5% CO₂. These updates can be found on page 22, line 9 of the revised manuscript.

Minor concerns:

1. P3, lines 2-3, this statement needs citation.

The appropriate citation has now been added to support this statement in P3, Line 3.

2. P3, line 24, "Excitation-Inhibition (E/I) balance".

We thank the reviewer for their careful evaluation. This term has been corrected as requested in P3 line 24.

3. P4, lines 5-7, I wouldn't say that the prominent E/I balance theory is the balance of AMPA EPSCs and GABA IPSCs. The consensus in the field, including the point discussed in Turrigiano 2012, is that E/I balance results from the coordination of multiple excitatory and inhibitory elements. AMPARs may have been studied more extensively than other channels (e.g. NMDARs), but that doesn't mean it's the only excitatory player here. Similar issues in discussion (P12, lines 18-19): it's more accurate to say, "a new molecular mechanism that contributes to E/I balance".

We thank the reviewer for this important and insightful comment. We agree that the E/I balance theory encompasses the coordination of multiple excitatory and inhibitory elements, not solely AMPA and GABA currents. In line with the reviewer's suggestion, we have revised the manuscript to reflect this broader and more accurate interpretation.

Specifically, on page 4, line 6, the text has been revised to: *“These data appear to be at odds with the prominent E/I balance theory, which posits that AMPA EPSCs—more extensively studied than other channels such as NMDARs—and GABA IPSCs are balanced in neurons for optimal neuronal function”*

Additionally, in the discussion section (page 13, line 23), we now state: *“We have discovered a new molecular mechanism that contributes to E/I balance.”*

These changes ensure that our description of E/I balance aligns with the current consensus in the field, including the perspective articulated in Turrigiano (2012). We appreciate the reviewer’s guidance in improving the accuracy and clarity of our manuscript.

4. P4, line 13, "whether NMDAR-mediated activity (or transmission) is regulated by ..."

This sentence has been modified as suggested in P4, line 14.

5. The two traces in Fig S1 seem to have different scales on Y axis (baseline), are these recordings obtained under similar settings? The legend is a bit ambiguous; are traces shown here the average waveforms of recordings from all 6 cells, or are they just representative traces from a given cell?

We have replaced the traces in Figure EV1 (previously S1) to ensure consistent Y-axis scales and clarified in the legend. Additionally, quantitative analyses have been included alongside the traces to support the visual data.

6. P4, line 28, "the mice were perfused and their brain sections were ..."

The sentence has been revised for clarity as suggested in P4, line 29.

7. In Fig 2B, could the authors explain why NMDAR-EPSCs do not differ between WT and KO at low intensities? Would NMDA/AMPA ratios also be similar within this range? Does this indicate that this effect on NMDAR transmission is input-dependent?

We appreciate this thoughtful question. In response, we have added additional recordings for NMDAR-EPSCs for the input-output curve analysis. The updated NMDAR-EPSC data were analyzed using a two-way repeated-measures ANOVA, followed by Sidak’s multiple comparisons. The results show that NMDAR-EPSC amplitudes are significantly reduced in $\beta 1-3$ -CRISPR neurons at stimulation intensities of 30, 40, 50, and 60 μ A, but not at lower intensities. Although the exact reason is currently unknown, it is possible that a threshold-dependent or nonlinear recruitment of NMDAR-mediated responses could occur in the $\beta 1-3$ -CRISPR condition.

In Fig 2B, NMDAR-EPSCs were recorded in the presence of the AMPAR antagonist CNQX, and AMPAR-EPSCs were recorded with the NMDAR antagonist APV. Due to independent acquisition, we could not calculate NMDA/AMPA ratios from the same cells. No significant differences were found in the AMPAR-EPSC input-output curves between groups. These clarifications and updated data are reflected in the revised text (p35, line 17) and figure 2B and C.

8. Could the authors explain why data in Fig 2B are subjected to two-way ANOVA tests? It appears that only the amplitudes at the highest intensity are compared, what's the second factor here? If the input intensity is treated as the second factor, then a two-way ANOVA for repeated measurements should be used. If so then this should be clarified in the text or legend. (Same for PPRs in Fig 2E)

Thank you for pointing this out. We used two-way ANOVA because both group (WT vs. KO) and input intensity were considered as factors in the analysis of Figure 2C. As these measurements were taken across increasing stimulation intensities within the same cells, a repeated-measures two-way ANOVA was applied. We have now clarified this in the figure legend on page 35, line 19. A similar clarification has also been added to the legend for Figure 2E, as paired-pulse ratio (PPR) data were analyzed using the same statistical approach.

9. Fig 3F is not called in the text.

Figure 3F has now changed to Figure 4D, referenced in the main text in P8 line24.

10. Fig S3A is not called in the text; the legend indicates these are done in HEK cells, as mentioned in above, could the authors provide rationales for using HEK and/or neuronal culture?

We thank the reviewer for pointing this out. Figure S3A is now cited in the main text (Page 6, line 21, currently referred to as Figure EV4A). HEK293 cells were initially used due to their high transfection efficiency, which facilitated the evaluation of plasmid expression and construct functionality. However, to better approximate physiological conditions and validate our findings in a more relevant neuronal context, we subsequently performed parallel experiments in primary neuronal cultures. This rationale has been clarified in the revised manuscript (Page 6, line 21).

11. Data in Fig S4B are similar to those in Fig 2B (both are illustrating repeated input-output measurements), could the authors explain why they choose to use an unpaired t-test here (rather than a two-way ANOVA)?

We appreciate the reviewer's suggestion regarding the appropriate statistical analysis. In response to the comment, we have reanalyzed the data using two-way repeated-measures ANOVA, which is more appropriate for assessing the influence of both group and stimulation intensity. The updated analysis and figure legend have been incorporated in the revised Figure EV5B.

12. In Fig S5A, if the authors intend to show the same current injection can evoke more spikes in certain conditions, then they should pick traces evoked by the same current step. Also it is not clear what parameter is being compared in Fig S5B.

We agree with the reviewer's comment and have replaced the traces in Fig S5A (current Figure 4A) to ensure they are from the same current injection step, thereby providing a more accurate comparison. Additionally, we have revised the figure legend for Fig S5B (current Figure 4A) to specify the parameter being quantified—the number of action potentials evoked at each current injection step—and to improve clarity overall (P36, line18).

13. In methods (P19, line 15), it's curious that the authors perform all recordings at 28C rather than at physiological temperatures, is there a specific reason for this choice? It's been reported that both the amplitude and frequency of NMDAR mEPSCs are temperature-sensitive (PMID: 18068304), showing substantial differences between 25C and 35C. Could some of the low frequencies in Fig 3F be caused by the recording temperature?

We thank the reviewer for this insightful comment. Initially, we tested various recording temperatures and found that temperatures above 30°C often led to reduced viability of the brain slices. Ultimately, we chose to perform recordings at approximately 28°C±2°C as a compromise—it allowed for extended slice stability and reduced metabolic stress, while remaining closer to physiological conditions compared to room temperature. We acknowledge that recording temperature can influence NMDAR mEPSC properties; however, all experiments were conducted under identical conditions, allowing for valid comparisons between experimental groups. This has been updated in the Methods section of the manuscript (Page 20 line 15).

14. Also in methods (P19, line 18), the authors state that data were filtered at 2 kHz and digitized at 10 kHz, does this include the evoked EPSCs (e.g. Fig 2)? I'm a bit concerned that the fast EPSC rising phase cannot be fully captured with a 2 kHz filter (if digitized at 10 k then the filter band should not be lower than 5k).

We thank the reviewer for highlighting this important technical detail. Upon reviewing our data acquisition parameters, we found that EPSC signals were actually filtered at 1 kHz and digitized at 10 kHz. We acknowledge that this filtering bandwidth may indeed limit the resolution of fast-rising components of EPSCs, and we appreciate the reviewer's attention to this issue.

To further address this concern, we conducted additional paired-pulse ratio (PPR) analyses in multiple cells. These analyses revealed no significant differences in PPR values between groups, supporting the robustness of our synaptic transmission findings despite the filtering limitations.

We have now updated the Methods section (Page 20, line 18) and the Figure 2E legend to reflect the accurate acquisition parameters and the inclusion of these additional PPR data.

Referee #3:

1. The term "single cell" used by the authors is confusing. It appears that "single cell" actually refers to a "single cell type," specifically CA1 pyramidal neurons. Clarifying this terminology would improve readability and accuracy.

We appreciate the reviewer's insightful observation regarding the term "single cell." To avoid confusion, we have revised the text to explicitly state "single-cell level" or "hippocampal CA1 neurons," as seen in p6, line 6; page 12, line 1 and line 20.

2. In Figure 1E, the rightmost graph shows the amplitude of mIPSCs despite the frequency of mIPSCs being nearly abolished. Could this discrepancy be due to incomplete knockout efficiency?

The observed residual mIPSC amplitude (~10 pA) in Figure 1E likely reflects the remaining few detectable events. While our CRISPR-mediated knockout system is highly efficient, no gene-editing tool achieves absolute completeness. However, the residual amplitude is unlikely to result from incomplete knockout for several reasons:

1. The near-total reduction in mIPSC frequency (<5% of control) strongly supports the effectiveness of the knockout.
2. The residual amplitude of 10 pA is consistent with the baseline noise level of our electrophysiological system. For this kind of genetic manipulation, even there is only one mIPSC event remaining in the CRISPR neurons, there is an amplitude number. This remaining mIPSC could be due to incomplete degradation of remaining beta subunit of GABA_A receptors in the neuron.
3. By targeting β 1-3 subunits, we have demonstrated that elimination of GABA_AR in individual hippocampal CA1 pyramidal neurons led to the silencing of GABAergic transmission (mIPSCs and sIPSCs). The efficacy of the GABA_AR KO vector was previously validated through immunostaining and electrophysiology in primary cultured neurons, which confirmed the loss of β 1, β 2, and β 3 subunit expression and the absence of mIPSCs (Duan et al., 2019). In the present study, we further show that deletion of these β subunits abolished GABA-evoked whole-cell currents, reinforcing the specificity and effectiveness of the KO strategy.

We thank the reviewer for this thoughtful observation. In response, we have revised the manuscript by replacing the word "completely" with "essentially" to more accurately reflect the observed effects. Additionally, we have clarified the knockout efficiency in the revised manuscript on page 11, line 6.

3. In Figure 2C, the authors state that "AMPA-mediated EPSCs were slightly but not significantly decreased compared to control neurons." However, the data show a notable reduction in AMPAR-EPSCs in the B123-CRISPR condition. Additionally, the authors employed a

two-way ANOVA for statistical analysis, but a repeated measures two-way ANOVA would be more appropriate for these experiments. Reanalyzing the data using the correct statistical method is recommended.

We thank the reviewer for the helpful comments and for highlighting the need for appropriate statistical analysis.

To clarify the earlier confusion, and to maintain a consistent number of samples for the AMPAR-EPSC input-output curve analysis, we have added additional recordings for NMDAR-EPSCs. The updated NMDAR-EPSC analysis confirms a significant reduction in β 123-CRISPR neurons compared to wild-type controls at stimulation intensities of 30 μ A (* $p < 0.05$), 40 μ A, 50 μ A, and 60 μ A (**** $p < 0.0001$).

Regarding AMPAR-EPSCs, we reanalyzed the data using repeated-measures two-way ANOVA followed by Sidak's multiple comparisons at individual stimulation intensities. This analysis revealed no statistically significant differences between the β 123-CRISPR and control groups ($p = 0.21$ – 0.99), and the overall group effect was also not significant ($p = 0.4661$). The corresponding source data have also been uploaded for verification.

Consistent with this, our experiments—including mEPSC recordings and AMPA-evoked whole-cell current measurements (see revised Figures EV2 and EV4B)—also showed no significant changes in AMPAR-mediated transmission. These findings further support the conclusion that AMPAR function remains largely unaffected by β 1–3-CRISPR. These updates have been incorporated into the revised manuscript, including the updated Figures 2B and 2C, the corresponding figure legend (p. 35, line 20), and the Methods section (p. 26, line 12).

4. In Figure 2D, the representative traces for β 123-CRISPR neurons display dual synaptic inputs to the cell. It is unclear whether dual-synaptic traces are suitable for the analysis presented. The authors should justify the inclusion of these traces or reconsider their use.

We thank the reviewer for this valuable comment. To address this concern, we have replaced the original traces in Figure 2D with new traces that more accurately reflect the data. Additionally, we have included additional PPR recordings and conducted new analyses, which are now included in the updated Figure 2E.

5. In Figure 3D, only the Control group exhibited a washout effect. Did the sgRNA-resistant GABAARs used for recovery display different kinetics compared to native GABAARs? If so, this difference should be addressed.

We appreciate the reviewer's insightful suggestion. While it would be interesting to analyze the kinetics of NMDAR-evoked whole-cell currents, our current methodology limits this comparison. Specifically, the slow solution exchange system we used (Picospritzer) does not allow for precise

kinetic measurements. Given these limitations, we have opted not to include the kinetics data in the manuscript. However, we are happy to include these results upon the reviewer's request.

6. While the authors convincingly demonstrate that reduced NMDARs impair SC-CA1 LTP and its associated memory behavior, the connection between the lack of GABA_ARs and decreased NMDAR function remains unclear. Although Figure 3 shows a GABA_AR recovery experiment, this only establishes correlations, not mechanisms. What is the putative mechanism underlying this phenomenon? Further investigation or discussion is warranted.

We appreciate the reviewer's comment regarding the need to clarify the mechanistic link between the lack of GABA_ARs and the decreased function of NMDARs. Below, we propose a putative mechanism based on the known roles of N-glycosylation and ER-associated degradation (ERAD) in regulating NMDAR expression and function:

The presence of high-mannose glycans on NMDAR-GluN1 subunits makes them highly susceptible to ubiquitination and endoplasmic reticulum-associated degradation (ERAD), enabling the activity-dependent ubiquitin-proteasome pathway to mediate the synaptic removal of NMDARs, thereby affecting their expression levels and currents (Ehlers, 2003, Kato et al., 2005, Scott and Panin, 2014). However, N-glycosylation is not universally required for AMPAR function (Scott and Panin, 2014), and the activity-dependent ubiquitin-proteasome pathway appears to have a differential role in regulating AMPARs (Widagdo et al., 2017). While GluA1 ubiquitination was suggested to signal AMPAR endocytosis(Widagdo et al., 2017), other studies disputed this, showing that neither GluA1 nor GluA2 ubiquitination regulates AMPAR surface expression or agonist-induced endocytosis (Lussier et al., 2011, Widagdo et al., 2015). Based on these findings, we propose a hypothesis in which the absence of GABA_ARs leads to elevated neuronal excitability, which in turn upregulates ERAD activity. This results in the degradation of improperly folded or glycosylated GluN1 subunits, thereby reducing NMDAR surface expression and function. This speculation provides a plausible explanation for the observed NMDAR deficits in the absence of GABA_ARs and suggests a potential mechanism through which GABA_ARs differentially influence synaptic NMDARs and AMPARs. Further investigation is needed to determine whether this mechanism accounts for the selective relationship observed between GABA_ARs and NMDARs. This has been discussed in the discussion section (p13, line 13).

7. In Figure 5B, the representative trace of B123-CRISPR mice in the Morris water maze suggests they follow the borders of the maze. Did all B123-CRISPR mice exhibit a similar behavior, or is this trace an isolated example? Providing more comprehensive data would strengthen this conclusion.

Not all the animals displayed the same behavior of following the maze borders; approximately 2 out of 9 exhibited this pattern. Therefore, we have replaced the sample trace with a more representative example. Additionally, we have replaced the swimming speed data with the total distance traveled data in the revised Figure 5F. Furthermore, we have updated Figure 5 and E to show both the number of crossings and the time spent in the target quadrant during the probe test ($p = 0.0013$).

8. The authors have inconsistently presented their data-some graphs include individual data points, while others display only mean {plus minus} SEM values. What is the rationale behind this discrepancy? Uniform representation of data would enhance clarity and transparency.

We appreciate the reviewer's observation regarding inconsistencies in data presentation. The discrepancy arose from the use of different versions of GraphPad at various stages of the study. In response to this suggestion, we have updated all relevant figures to consistently include individual data points alongside mean \pm SEM values, thereby improving clarity and transparency across the manuscript.

9. The authors used DCS to assess whether partial activation of NMDARs could rescue impaired working memory. Did the DCS treatment also include mice subjected to the Morris water maze test? Clarifying this experimental design is important for interpretation.

We appreciate the reviewer's suggestion to investigate the effects of D-cycloserine (DCS) on spatial memory using the Morris water maze (MWM) test. Previous studies, such as Portero-Tresserra et al. (*Psychopharmacology*, 2018, 235:1463–1477, <https://doi.org/10.1007/s00213-018-4858-z>), have shown that intra-hippocampal DCS administered 20 minutes before each of the five acquisition sessions (days) can rescue impaired spatial learning in the MWM in aged rats. However, most investigations of DCS employ single low-dose injections in short-term memory paradigms.

In our study, acute DCS application restored LTP in β 123-CRISPR neurons to levels comparable to controls. The Y maze spontaneous alternation assay requires a functionally intact dorsal hippocampus and has also been widely used to test spatial working memory. Given its sensitivity to hippocampal dysfunction and compatibility with the short-term pharmacological effects of DCS, we selected the Y-maze as an alternative approach to evaluate hippocampal function in CRISPR β 1-3 mice. While we agree that testing DCS in the MWM would be informative, we did not administer DCS to MWM-tested mice due to resource constraints and concerns about long-term DCS safety. This has been clarified on page 10, line 12.

Reference

1. Ehlers, M.D. 2003. Activity level controls postsynaptic composition and signaling via the ubiquitin-proteasome system. *Nat. Neurosci.* 6: 231-42. <https://doi.org/10.1038/nn1013>
2. Kato, A., Rouach, N., Nicoll, R.A., Brecht, D.S. 2005. Activity-dependent NMDA receptor degradation mediated by retrotranslocation and ubiquitination. *Proc. Natl. Acad. Sci. USA.* 102: 5600-5605. <https://doi.org/10.1073/pnas.0501769102>
3. Lussier, M. P., Nasu-Nishimura, Y., and Roche, K. W. 2011. Activity-dependent ubiquitination of the AMPA receptor subunit GluA2. *J. Neurosci.* 31: 3077-3081. <https://doi.org/10.1523/jneurosci.5944-10.2011>
4. Mu, Y., Otsuka, T., Horton, A. C., Scott, D. B., Ehlers, M. D. 2003. Activity-dependent mRNA splicing controls ER export and synaptic delivery of NMDA receptors. *Neuron*, 40, 581-94. [https://doi.org/10.1016/s0896-6273\(03\)00676-7](https://doi.org/10.1016/s0896-6273(03)00676-7)
5. Scott, H., Panin, V. M. 2014. N-glycosylation in regulation of the nervous system. *Adv Neurobiol*, 9, 367-94. https://doi.org/10.1007/978-1-4939-1154-7_17
6. Widagdo, J., Chai, Y. J., Ridder, M. C., Chau, Y. Q., Johnson, R. C., Sah, P., Huganir, R.L., Anggono, V. 2015. Activity-dependent ubiquitination of GluA1 and GluA2 regulates AMPA receptor intracellular sorting and degradation. *Cell Rep.* 10:783-795. <https://doi.org/10.1016/j.celrep.2015.01.015>

Dear Dr. Duan,

Thank you for the submission of your revised manuscript. We have now received the enclosed report from referee 2 who was asked to assess it. Unfortunately, referee 3 was not available to re-review your work, and I therefore asked referee 2 to please assess your response to all referee comments. I am happy to say that referee 2 is satisfied with your replies, and only raises a few minor issues that I would like you to address and incorporate before we can proceed with the official acceptance of your manuscript.

A few editorial requests will also need to be addressed:

- Please remove the figures from the ms file; the figure legends (main and EV) need to stay at the end of the ms.
- Please reduce the number of keywords to 5.
- Please correct the conflict of interest subheading to "Disclosure and Competing Interests Statement"
- Please remove the author credits from the ms file.
- The REFERENCE format needs to be corrected: - et al needs to be used after 10 author names; DOIs should only be used for preprints and datasets that have not been published yet. Please use the EMBO reports reference style.
- This FUNDING INFO is missing in our ms online submission system: Advanced Medical Technology Center, The First Affiliated Hospital, Zhongshan School of Medicine, Sun Yat-sen University. Please add.
- Please remove the SYNOPSIS IMAGE from the ms file and upload it separately (550 pixels wide x 200-600 pixels high). EMBO press papers are accompanied online by A) a short (1-2 sentences) summary of the findings and their significance, B) 2-3 bullet points highlighting key results and C) a synopsis image that is exactly 550 pixels wide and 200-600 pixels high (the height is variable). The synopsis image should provide a sketch of the major findings, like a graphical abstract. Please note that text needs to be readable at the final size. Please send us this information along with the final manuscript.
- Please remove the Reagents & Tools table from the ms file and upload it separately.
- Materials & Methods should be Methods
- BioRender should be acknowledged at the end of the Methods section in the following way:
Graphics:
(some of the... OR Figure #... OR synopsis) Graphics were created with BioRender.com.

Figure Legends - Comments

- Please note that the exact p values are not provided in the legends of figures 1E, 2B, 3C, D; 4B, D; 6C, EV3, EV4 C; EV5 B, C. Please provide exact values as reasonable.
- Please indicate the statistical test used for data analysis in the legends of figures 5E, EV1, EV3
- Please note that information related to n is missing in the legends of figures 2B, C, E; 4B, 5A, 6A, C, D
- Please note that the error bars are not defined in the legends of figures 1E; 2A, B, C, E; 3C, D; 4B, D; 5A, D, E, F; 6A, C, D; EV1, EV2, EV3, EV4B, C; EV5 C

I would like to suggest some minor changes to the abstract that needs to be written in present tense:

GABAA receptors (GABAARs) mediate most synaptic inhibition in the brain, but their cell-autonomous role in regulating glutamatergic transmission remains poorly understood. By targeting GABAAR β 1-3 subunit alleles (GABRB1-3) at once, we genetically eliminated GABAARs in individual hippocampal CA1 pyramidal neurons. We find that single-cell silencing of GABAergic transmission does not alter AMPAR-mediated synaptic transmission, but leads to a reduction in NMDAR-mediated synaptic transmission, loss of long-term potentiation (LTP), and impaired spatial memory. Genetic rescue experiments reveal that NMDAR-mediated whole-cell currents and synaptic transmission depend on specific GABAAR subtypes and are tightly regulated by neuronal excitability. Pharmacologically restoring NMDAR function in β 123-CRISPR mice rescues both LTP and spatial memory deficits induced by the loss of GABAARs in CA1 neurons. Our data uncover a previously unknown regulation of synaptic NMDAR functions by GABAARs at the single-cell level and provide insight into excitation and inhibition balance between GABAARs and NMDARs in the brain.

Referee #2:

The authors have addressed all my concerns and significantly strengthened the study in this revision. I especially appreciate the additional validation of previously questionable results and the revised discussion that provides important context for interpretation. In summary, this valuable study establishes a causal relationship between GABAAR function and NMDAR-mediated transmission in CA1 pyramidal neurons, revealing an important cell-autonomous mechanism underlying the maintenance of overall E-I balance.

Assessment of comments from other referees:

Referee 1:

Most comments are minor critiques such as typos, and I have no objection to these textual changes. The last comment regarding incomplete statistical analyses looks valid to me, and the authors have adequately addressed this issue by performing one-way ANOVA with Bonferroni corrections.

Referee 3:

1. The authors have addressed this comment sufficiently.
2. R3 is questioning whether the inconsistent reductions in mIPSC frequency and amplitude are due to incomplete GABAR knockout. I don't find these results unusual, as the authors have pointed out that even a single mIPSC event could have a higher-than-average amplitude, thus low frequency does not necessarily correspond to low amplitude. Since knockout efficiency is unlikely to be 100%, some residual events are normal. The authors have addressed this sufficiently.
3. While I agree with R3 that AMPAR-mediated EPSCs do look separated in Fig 2C, the differences are clearly much smaller than NMDAR-mediated ones. The authors state that these conditions are not significantly different after repeated two-way ANOVA test, which I find sufficient. As AMPAR-mediated EPSCs can be quite variable, it's not uncommon for groups with different means by eye to not reach statistical significance. The authors have also provided additional data in Figs EV2 and EV4 to show that the effects on AMPAR-EPSCs are minimal. I find these responses sufficient.
4. I agree with R3 that dual synaptic inputs are not suitable for PPR analysis. The authors have added new data for PPR recordings, which I believe do not contain these dual synaptic inputs? If so, then I find this response sufficient.
5. I agree with R3 that this difference in decay kinetics upon wash-out may indicate changes in GABAR kinetics after rescue experiments. However, I don't think this is unusual- the authors did these rescue experiments with individual GABAR subunits, which could very likely lead to different subunit compositions from those in WT neurons. As I don't think kinetics changes would affect the main conclusion of this study, a brief discussion of this possibility in the discussion would be sufficient.
6. R3 is asking about additional molecular mechanisms underlying the reduced NMDAR function following GABAR knockout, which I believe the editor has advised against. Therefore, although this is an intriguing question and definitely worth further investigation, I don't believe it is required for the current study. The authors have provided some fascinating speculations in the discussion, and I think this is sufficient.
7. Details of these behavioral assays are outside my expertise, so I will trust R3's comment on this issue. The authors' response looks sufficient to me.
8. I agree with R3 that data presentations should be consistent. The authors have addressed this issue sufficiently.
9. Again, which behavioral assay is the best for DCS application is outside my expertise. In my opinion, the authors have chosen a well-established assay and the results are convincing. I have no critique on authors' response to this comment.

All editorial and formatting issues were resolved by the authors.

Dr. Jing duan
Sun Yat-sen University, Guangzhou 510080, China
Neuroscience and Anatomy
China

Dear Dr. Duan,

I am very pleased to accept your manuscript for publication in the next available issue of EMBO reports. Thank you for your contribution to our journal.

Yours sincerely,
